# Biologically inspired microlens array camera for high-resolution wide field-of-view imaging

Jae-Myeong Kwon [1,2], Yejoon Kwon [3], Young-Gil Cha [1,2], Dong Hyun Han[1], Hyun-Kyung Kim [1,2], Je-Kyun Park [1], Min H. Kim [3] ✉ & Ki-Hun Jeong [1,2] ✉

Natural vision employs diverse strategies to achieve wide field-of-view imaging critical for environmental awareness. Here we report spatially offset ellipsoidal microlens array camera, inspired by the angular sampling strategy of *Xenos peckii* for high-resolution wide field-of-view imaging. The camera features optical units with spatially offset-coupled apertures and ellipsoidal microlenses onto a single planar sensor within a 0.94 mm total track length. Direction-specific spatial offsets and asymmetric microlens curvatures substantially reduce aberrations across a 140° field-of-view. Digital calibration and image stitching reconstruct complex surfaces such as microfluidic channels, dental phantoms, and human faces, producing one-megapixel images with 1.1-pixel error. This ultrathin camera provides high-resolution and wide field-of-view imaging of real-world targets in confined spaces for applications in machine vision, mobile imaging, and healthcare monitoring.

Natural vision exhibits remarkable multifunctionality and allows detection of motion, perceive depth[1,2], recognize color[3,4], and adapt to dynamic environments[5,6]. Among these, wide field-of-view (FOV) capability serves as a key mechanism for detecting predators and monitoring the environment[7–9]. The compound eyes of arthropods—such as dragonflies[10], fiddler crabs[11,12], and mantis shrimp[13]—utilize an array of ommatidia, each comprising a facet lens, crystalline cone, and photoreceptor cells. Their spherical arrangement provides panoramic vision by collecting light from various directions, while the narrow acceptance angles of individual ommatidia help suppress optical aberrations[14,15]. This optical configuration naturally results in low-resolution mosaic vision since each ommatidium functions as a single pixel[16]. Alternatively, jumping spiders rely on multiple camera-type eyes, which combine individual images from spatially distributed principal and secondary eyes[17–19]. Modular specialization among the eyes supports high-resolution central vision alongside peripheral motion detection. In addition, raptors[20,21] and chameleons[22] achieve a wide FOV using only two eyes, rather than multiple visual units. Both eyes are positioned laterally with minimal overlap or independent

control to support active scanning. Effective panoramic coverage results from these simple optical strategies, though they involve trade-offs between structural simplicity and spatial resolution. Such natural adaptations inspire the strategic design of compact wide FOV imaging systems that balance optical simplicity with functional versatility.

Biological inspiration for wide FOV imaging drives the development of artificial compound eye (ACE) cameras that emulate the structure and function of natural vision systems. ACE cameras often use spherically arranged microlens arrays (MLAs) on curved photodetector arrays[23–25]. The spatial resolution of ACE cameras is intrinsically limited by the number of microlens–photodetector pairs, each forming a unit of the resulting mosaic image. This configuration provides geometric simplicity over traditional wide-angle lenses and mechanically scanning systems. Other approaches integrate spherically arranged MLAs on conventional planar sensors using optical waveguides[26,27], extended depth-of-focus microlenses[28], or additional lens sets[29,30]. Despite improvements in spatial resolution, current implementations face challenges in further miniaturization and mass production due to complexities in fabrication and integration. Unlike

[1]Department of Bio and Brain Engineering, Korea Advanced Institute of Science and Technology (KAIST), Daejeon, Republic of Korea. [2]KAIST Institute for Health Science and Technology (KIHST), Korea Advanced Institute of Science and Technology (KAIST), Daejeon, Republic of Korea. [3]School of Computing, Korea Advanced Institute of Science and Technology (KAIST), Daejeon, Republic of Korea. ✉e-mail: minhkim@kaist.ac.kr; kjeong@kaist.ac.kr

curved wide FOV imaging systems, metalens[31–34] and lensless[35–37] cameras recently serve as ultrathin alternatives based on wavefront manipulation or intelligent image processing. Both approaches still suffer from nanofabrication challenges, limited operational bandwidth, or high reconstruction cost. In contrast, planar MLAs on image sensors provide ultrathin and scalable cameras with insect-inspired imaging for high-speed capture[38], high dynamic range[39], and depth estimation[40]. However, planar MLA cameras still struggle with high-resolution wide FOV imaging due to optical aberrations at wide angles[41]. Conventional spherical microlenses induce significant astigmatism under oblique illumination, producing blurred point spread functions (PSFs) due to sagittal–tangential focal mismatch.

Here we report spatially offset ellipsoidal microlens array (SOEMLA) camera for low-aberration wide FOV imaging, inspired by the vision scheme of *Xenos peckii*. The parasitic insect *Xenos peckii* features a unique visual system of dozens of eyelets blending compound eyes and camera-type eye traits (Fig. 1a)[42–44]. Each eyelet, containing a single lens on hundreds of photoreceptors, captures an individual image from a distinct direction, and together, they integrate partial views into a high-resolution wide FOV image. The SOEMLA camera utilizes spatially offset-coupled apertures (SOAs) and ellipsoidal microlenses on planar CMOS image sensor arrays (CMOS ISA) (Fig. 1b). Like *Xenos peckii*'s visual sampling, each optical unit contains SOAs, an ellipsoidal microlens, and a designated pixel region on the CMOS ISA, which captures a separate view from a particular direction. The captured array images are digitally corrected and stitched to reconstruct a wide FOV image. Large viewing angles in wide FOV imaging primarily cause field curvature, astigmatism, and distortion. In particular, the SOAs consist of upper and lower apertures that are laterally displaced by a spatial offset to define the angular range of each optical unit. The SOEMLA camera achieves seamless angular coverage across the wide FOV through spatially varying offsets of the aperture arrays. The SOAs also allow individual adjustment of the focal lengths to compensate for field curvature and align all viewing angles to a common focal plane on the planar CMOS image sensor. In addition, the SOEMLA camera employs ellipsoidal microlenses with asymmetric curvature along the sagittal and tangential planes. Unlike conventional spherical microlenses, ellipsoidal microlenses use angle-specific asymmetric curvatures to form sharp point-spread functions (PSFs) and thus substantially suppress astigmatism across the entire FOV. Finally, the SOEMLA camera produces low-aberration partial images that support accurate distortion correction and image stitching, leading to high-resolution wide FOV imaging in an ultrathin form factor.

## Results
### Microfabrication and packaging of SOEMLA camera
The SOEMLA camera was fabricated through wafer-level processing of the SOAs and ellipsoidal microlenses, and subsequently integrated onto commercial CMOS ISA (Fig. 2a). Chromium (Cr) evaporation and lift-off were applied to both sides of a 4-inch borosilicate glass wafer to define light-blocking regions for the SOAs (step I). Spatial offsets were introduced during SOAs fabrication to assign each optical unit with a unique orientation and viewing angle (Supplementary Figs. 1–3). Elliptic microcylinders were patterned using a thermoplastic photoresist (DNR-L4615D, Dongjin Semichem Co., Ltd.) (step II) and thermally reflowed to form ellipsoidal microlenses (step III). Note that precise control of the ellipsoidal microlens curvature was achieved by adjusting the lateral dimensions and aspect ratios of monolithically defined microcylinders with a uniform thickness (Supplementary Fig. 4). Finally, the microfabricated SOEMLAs were mounted onto the CMOS ISA (IMX477, SONY Corp.; 12.3 MP, unit pixel: $1.55 \times 1.55\ \mu m^2$) via precision flip-chip bonding using alumina spacers and UV-curable epoxy (step IV). Detailed fabrication parameters for each step (I–IV) are summarized in Supplementary Table 1. The optical image clearly demonstrates that the ellipsoidal microlenses are radially arranged with spatially varying dimensions and orientations, corresponding to the displaced and rotated $5 \times 7$ configurations designed to extend the overall FOV (Fig. 2b). In particular, the peripheral microlenses exhibit large aspect ratios and lens sizes to capture wide viewing angles (detailed parameters of each unit are provided in Supplementary Fig. 5). The surface profile of the ellipsoidal microlens was measured using confocal laser scanning microscopy (OLS5000-LAF, Olympus) (Fig. 2c). The measured profile closely matches the Zemax OpticStudio (Ansys, Inc.) target design, minimizing astigmatism and field curvature along both the X–X' and Y–Y' cross-sections. The radius of curvature (ROC) was calculated along the major and minor axes from center to periphery based on the measured surface profile (Fig. 2d). The ROC gradually increases from center to periphery, reflecting the design rule of assigning longer focal lengths to peripheral microlenses for off-axis field curvature compensation (Supplementary Fig. 6). The major-to-minor lens curvature ratio increases toward periphery for minimal astigmatism. The measured ROCs match well with ZEMAX calculations (average error: 0.88%) and demonstrate precise control via monolithic microfabrication. The fully packaged SOEMLA camera exhibits a highly compact form factor ($8.3 \times 6.7$ mm, total track length: 0.94 mm), comparable in size to a US penny (Fig. 2e and Supplementary Fig. 7).

### SOEMLA-based high-resolution wide field-of-view imaging
The SOEMLA camera provides high-resolution wide FOV imaging by suppressing optical aberrations and maintaining uniform image quality over a broad angular range. In particular, tuning the aspect ratio (AR; minor-to-major axis length) of the ellipsoidal microlenses balances the sagittal and tangential focal planes and minimizes astigmatic blur at oblique incidence. A SOEMLA camera with ARs ranging from 0.80 to 1.00 was fabricated to evaluate astigmatism correction at oblique incidence (Fig. 3a). Consequently, the minor-axis focal length was tuned while the major-axis focal length was kept constant at 800 μm, as demonstrated by the confocal sectioning results (Supplementary Fig. 8). The spacer-defined image distance of the camera packaging was set to match the major-axis focal length, which primarily influences tangential imaging performance. The 0.90-AR microlens produces sharp focus in both sagittal and tangential directions when evaluated at a 20° visual axis angle, confirming effective single-lens astigmatism correction. In contrast, the non-optimal-AR microlenses form sharp focus in the tangential direction but blurred sagittal features due to focal plane mismatch. MTF50 measurements at the same visual axis angle further quantify the variation in directional resolution with respect to ARs (Fig. 3b). All MTF50 measurements in this section were performed under identical imaging conditions (illuminance: 500 lux, ISO 100, image signal processing off, and exposure time individually adjusted to compensate for light throughput differences across lenses, working distances, and incidence angles). The AR of 0.90 yields the highest and most balanced MTF50 in both sagittal (S) and tangential (T) directions, while other values increase directional asymmetry caused by astigmatism. This optimal AR closely matches the theoretical value of 0.86 calculated by the geometric optical analysis. The experimental results demonstrate that ellipsoidal microlenses effectively reduce angular-dependent astigmatism without relying on complex optical assemblies or active control. Note that the 0.90-AR microlens corresponds to a 20° visual axis, while each optical channel can be tuned in curvature and asymmetry across the full FOV to match local incidence angles.

The SOEMLA camera corrects astigmatism effectively at various viewing angles by extending the single-unit optimization through tailored curvature of ellipsoidal microlenses (Supplementary Fig. 9). Ray-tracing results further show that the SOEMLA configuration maintains a nearly constant RMS spot radius across wide incidence angles, confirming superior aberration correction capability (Supplementary Figs. 10–11). The spatial resolution of the SOEMLA camera was

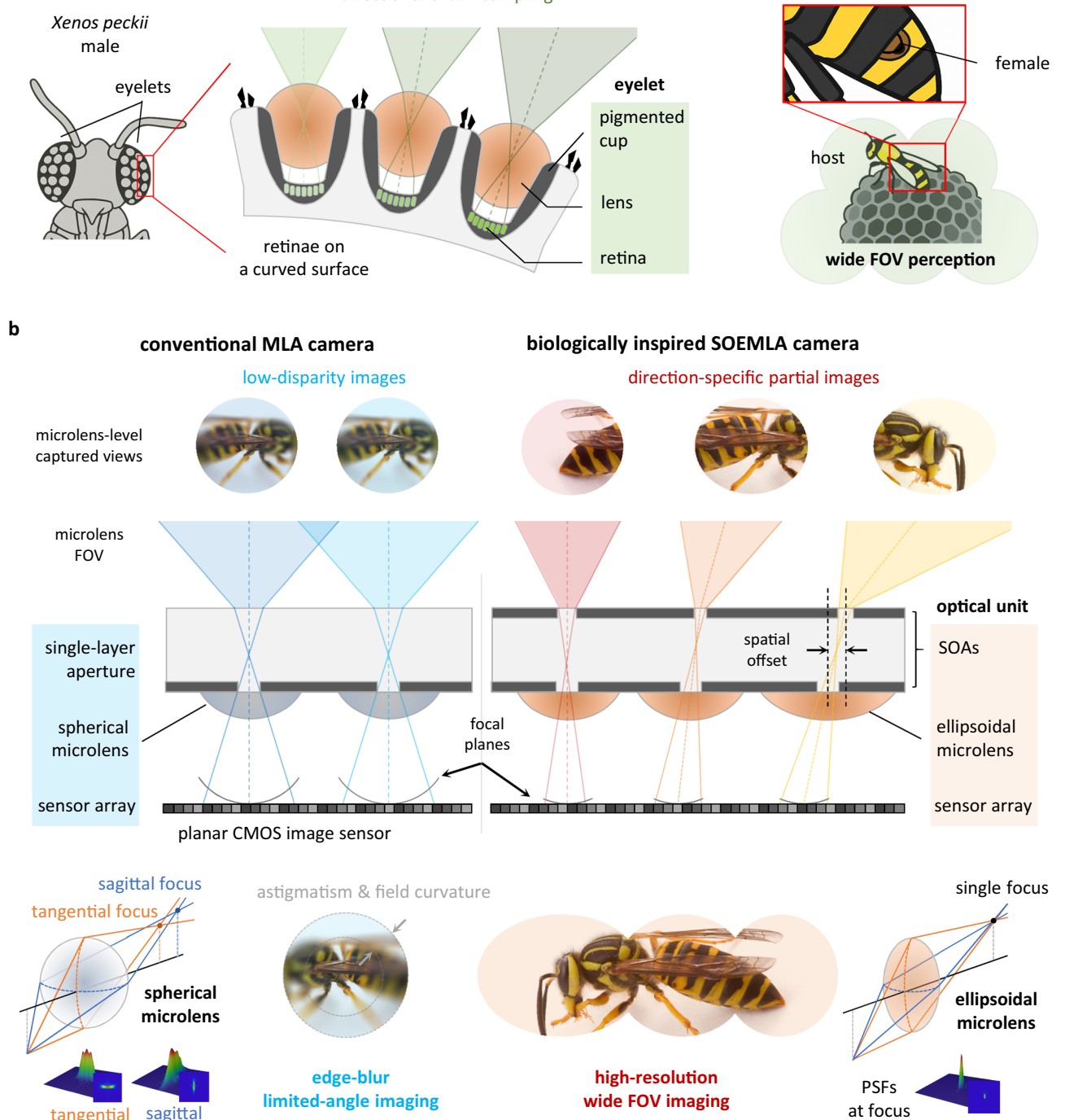

**Fig. 1 | Biologically inspired wide field-of-view (FOV) camera using spatially offset ellipsoidal microlens arrays (SOEMLAs). a** Unique visual strategy of *Xenos peckii*. The retinae of *Xenos peckii* are spherically arranged on a curved surface to achieve precise visual detection across a wide FOV via multi-angle chunk sampling. **b** Comparison between a conventional MLA camera and the SOEMLA camera. Conventional microlenses experience severe field curvature and astigmatism, along with a narrow FOV due to low disparities between optical units. In contrast, the SOEMLA camera, inspired by *Xenos peckii*, utilizes spatially offset-coupled apertures (SOAs) and ellipsoidal microlenses on a single planar CMOS image sensor. The SOAs with variable spatial offsets between upper and lower apertures effectively capture direction-specific partial images while minimizing field curvature. Ellipsoidal microlenses, integrated onto the lower apertures, form astigmatism-free PSFs across a wide FOV on the CMOS sensor.

evaluated with respect to the incidence angle and compared with other MLA cameras (Fig. 3c). Conventional MLA camera using spherical microlenses with single-layer apertures shows significant resolution degradation at large incidence angles due to both optical field curvature and astigmatism (blue). For a fair comparison, the spherical MLA camera was fabricated with the same optical specifications as the central optical unit of the SOEMLA but employed single-layer apertures ($f = 440\,\mu m$, $F/3$). The spatially offset microlens arrays (SOMLAs), specifically designed for comparison, adopt the SOAs design but still employ spherical microlenses. The SOMLA camera eliminates field curvature by adjusting the focal length of each microlens to the designated viewing angle. However, astigmatism remains uncorrected

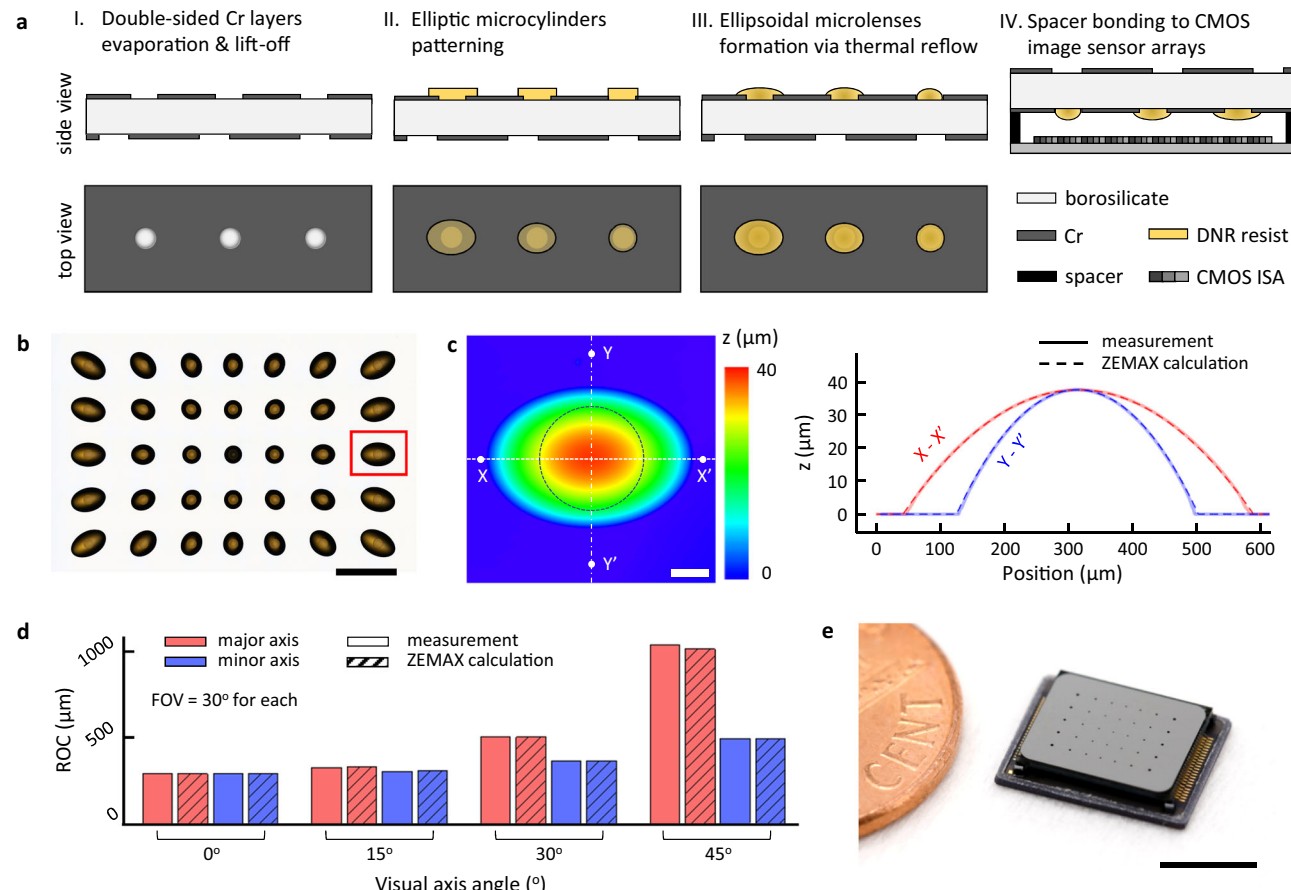

**Fig. 2 | Microfabrication and packaging of the SOEMLA camera.**
**a** Microfabrication steps of the SOEMLA camera. The SOAs were defined on both sides of a borosilicate glass wafer by using Cr evaporation and lift-off process (step I). Elliptic microcylinders were photolithographically defined and reflowed into ellipsoidal microlenses via resist melting (step II-III). Finally, the SOEMLAs were mounted onto a single CMOS ISA with alumina spacers (step IV). **b** Optical image of the microfabricated SOEMLAs with variable sizes and asymmetry. The ellipsoidal microlenses feature radially arranged curvatures with variable spatial offsets to reduce astigmatism and field curvature in wide FOV imaging. Scale bar: 1 mm. **c** 3D surface map of the ellipsoidal microlens highlighted by the red box in b, measured by confocal laser scanning profilometer. The black dashed line represents the size

of the central spherical microlens for reference (left). Comparison of the microfabricated ellipsoidal microlens profile with its optical design (right). Cross-sectional profiles along the major (X-X') and minor (Y-Y') axes show strong agreement between the measured data and the optical design. Scale bar: 100 μm.
**d** Radius of curvature (ROC) of the ellipsoidal microlenses. The ROC increases with the visual axis angle to correct the field curvature, while the difference between the ROCs of the major and minor axes diverges to reduce astigmatism. The measured ROC values align closely with the target values, indicating a high level of precision (average error: 0.88%) in microfabrication. **e** Captured image of the fully packaged SOEMLA camera. Scale bar: 5 mm.

and resolution improves only in either the sagittal or tangential direction depending on the spacer gap (green). In contrast, the SOEMLA camera exhibits consistently high and directionally balanced spatial resolution even at large incidence angles, achieving superior performance across the wide FOV of ±70° (red). The resolution gain of SOEMLA over the spherical reference increases exceptionally with incidence angle owing to directionally adaptive optical design. The SOEMLA camera clearly resolves line pairs across all angles, whereas the spherical system blurs in all directions and SOMLAs primarily in the tangential (Fig. 3d).

The SOEMLA camera features a compact form factor optimized for wide FOV imaging, distinct from commercial designs (Fig. 3e). The SOEMLA camera achieves 0.94 mm total track length (TTL) and 140° FOV with a 7.9 mm sensor, whereas the commercial camera (Camera Module 3 Wide, Raspberry Pi Ltd.) has 8.3 mm TTL and 120° FOV with a 7.4 mm sensor. In addition, the MTF50s of both cameras were evaluated across working distances (WDs) ranging from 10 to 100 mm, under both normal and 45° oblique incidence (Fig. 3f). The commercial camera shows strong depth dependence with resolution peaking sharply near the focal plane. The MTF50 peak values vary significantly in both directions under oblique incidence due to astigmatism. Note

that the lens position of the commercial camera was set for a WD of 50 mm for consistency. In contrast, the SOEMLA camera decouples depth-dependent focal variation from angular aberrations, ensuring consistent MTF50 values across both depth and angle. The SOEMLA camera exhibits a hyperfocal distance shorter than 10 mm, resulting in effectively infinite DOF (Supplementary Note 1). Unlike conventional lens systems that extend DOF at the cost of brightness, the SOEMLA camera maintains overall light collection efficiency through its inherently short focal length. Measured signal-to-noise ratio (SNR) demonstrates practical photon throughput across visual-axis angles (Supplementary Figs. 12–13). Along with high resolution and infinite DOF, the SOEMLA camera clearly demonstrates proximal imaging (< 50 mm), thereby reducing the overall optical path (WD + TTL) and facilitating further miniaturization. The experimental results consequently support the SOEMLA camera as a compact, depth-invariant platform for wide FOV and close-range imaging.

**Camera calibration and wide field-of-view image reconstruction**
Seamless wide FOV imaging with the SOEMLA camera is achieved through a combination of an MLAs calibration stage (step I-IV) and an image reconstruction stage (step V-VI) (Fig. 4a). The calibration stage

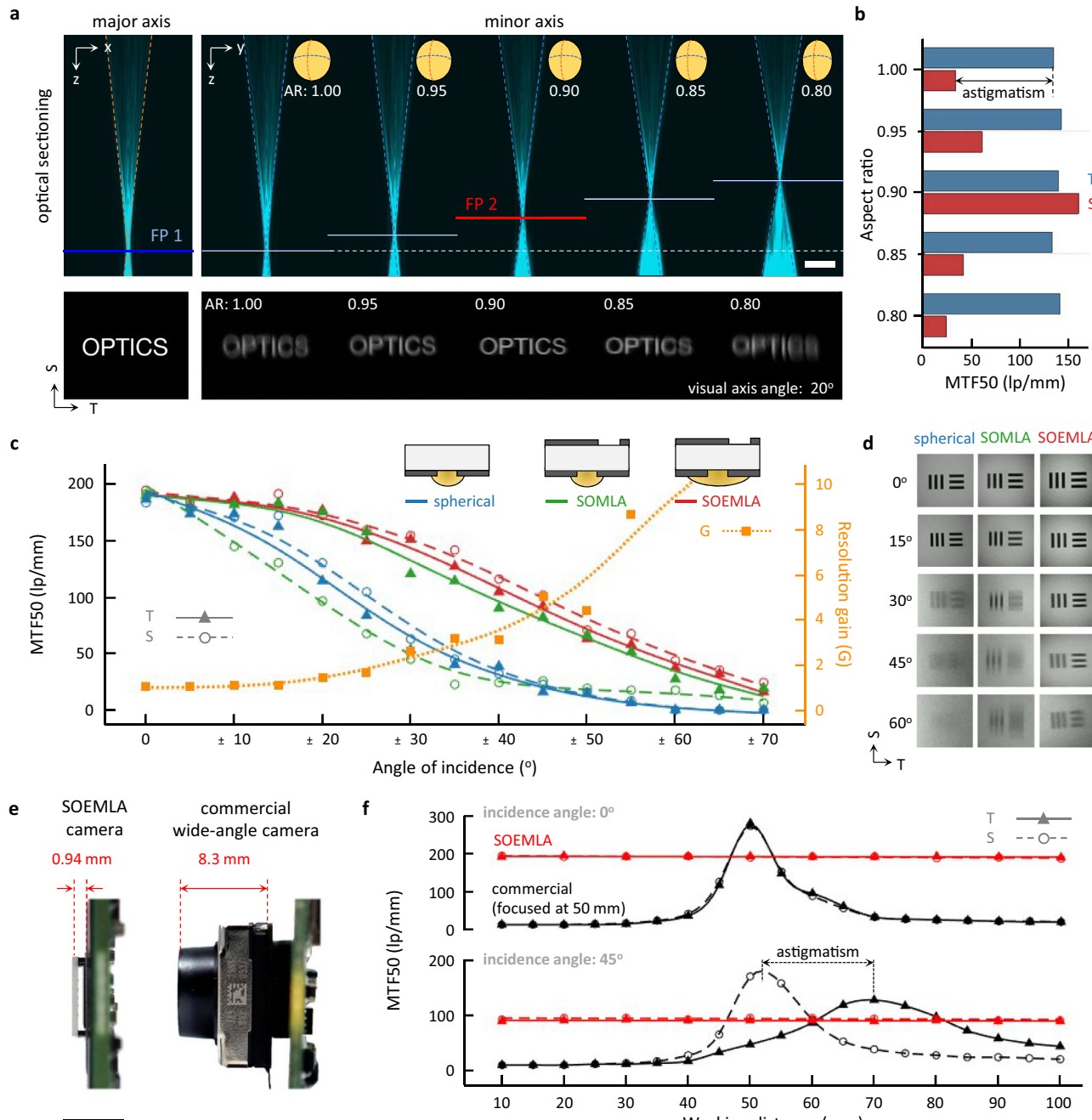

**Fig. 3 | Astigmatism-free and wide FOV imaging using the SOEMLA camera.**
**a** Astigmatism correction of a single ellipsoidal microlens. Optical sectioning of ellipsoidal microlenses with different aspect ratios (ARs). All microlenses share the same focal length along the major axis but have different focal lengths along the minor axis. Captured images show sharp focus in both tangential and sagittal directions with an ellipsoidal microlens AR of 0.90, closely matching the theoretical value of 0.86. Scale bar: 100 µm. **b** Measured MTF50 for the visual axis angle of 20° depending on the aspect ratio of ellipsoidal microlenses. The ellipsoidal microlens (AR: 0.90) significantly reduces astigmatism caused by the visual axis deviates further from the optical axis. T and S denote the tangential and sagittal directions. **c** Measured MTF50s and **d** captured line pair images at various incidence angles for

spherical microlens (blue), SOMLA (green), and SOEMLA (red) cameras. The SOEMLAs exhibit exceptionally high and balanced MTF50 values on both planes across the field. The resolution gain, i.e., the SOEMLA-to-spherical MTF50 ratio, increases sharply at large incidence angles. **e** Size comparison of SOEMLA camera (TTL: 0.94 mm, diagonal FOV: 140°) and a commercial compact wide-angle camera (TTL: 8.3 mm, diagonal FOV: 120°). Scale bar: 5 mm. **f** Measured MTF50 as a function of working distance for the SOEMLA and commercial cameras at incidence angles of 0° (top) and 45° (bottom). The commercial camera (focused at 50 mm) shows shallow DOF, while the SOEMLA camera maintains uniform resolution over a wide range without astigmatism.

proceeds in the order of microlens region-of-interest (ROI) detection, lens shading correction, distortion correction, and homography calculation. The reconstruction stage then uses the resulting calibration parameters to perform homography-based stitching and weighted blending. In more detail, the calibration process begins

with the automatic detection of each microlens's effective ROI from a white reference frame. Next, lens shading correction is applied to each partial image using pixel-wise gain factors derived from flat-field white images, compensating for the radial intensity fall-off specific to each optical unit. Radial and tangential distortions are

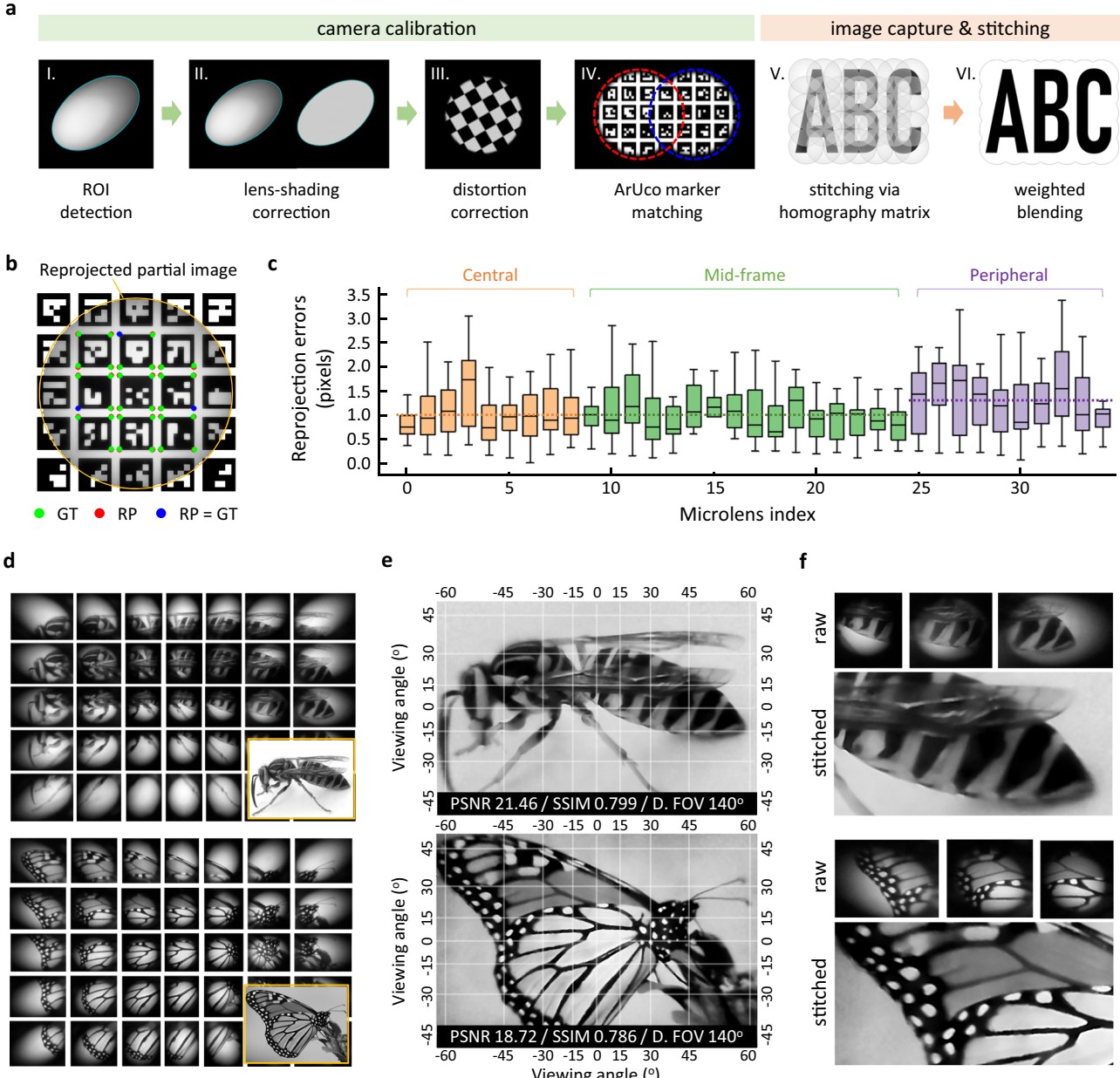

**Fig. 4 | SOEMLA camera calibration and image stitching using feature-based registration. a** Image processing procedure. The camera calibration allows for uniform brightness and distortion-free images while determining the warping matrix. After array image capture, distortion-corrected images are stitched via homography transformation (HT) with weighted blending. **b** Calculation of the reprojection error from the ArUco target's corner ground truth (GT, green) and reprojected points (RP, red) with overlaps in blue. The RP data from each partial image determines the warping matrix for HT. **c** Reprojection errors across the full FOV. Reprojection errors from the central ($\pm 0$–$30°$) to the mid-frame ($\pm 15$–$55°$) and peripheral ($\pm 30$–$70°$) regions consistently remain around 1.1–1.3 pixels, demonstrating high calibration accuracy and minimal aberration for precise image stitching. In the box plot, the center line indicates the median, the box represents

the 25th–75th percentiles, and the whiskers extend to $1.5 \times$ the interquartile range. Each microlens is spirally indexed from the center of $5 \times 7$ arrays. **d** Partial images captured by the SOEMLA camera of the wasp (top) and butterfly (bottom) displayed on an LCD screen. Each partial image corresponds to a different visual axis angle, with adjacent images redundantly overlapping for seamless stitching. The reference target images are displayed on the bottom right. **e** Fully stitched wide FOV images. The reconstructed images exhibit minimal aberration and closely match the original screen across the wide FOV (horizontal: 120°, vertical: 90°, diagonal: 140°). The peak signal-to-noise ratio (PSNR) and structural similarity index measure (SSIM) values confirm high reconstruction quality. **f** Seamless reconstruction from raw images. Magnified views of d and e demonstrate smooth integration, preserving fine details without mismatches.

then corrected by estimating distortion coefficients from checkerboard calibration images[45]. Finally, partial images are spatially aligned using an ArUco marker array designed for the SOEMLA layout. Corner points of detected markers are extracted using OpenCV's built-in functions, and the resulting homography matrix provides accurate registration across all views. Calibration is performed once prior to imaging, and the resulting parameters are reused for

subsequent reconstructions without additional recalibration. Following calibration, multi-view reconstruction is performed using homography-based stitching and Poisson blending. Each partial image is geometrically aligned using the precomputed warping matrix, and Poisson blending is applied to suppress gradient discontinuities along boundaries, yielding seamless integration[46]. The entire pipeline is implemented in Python using standard image-

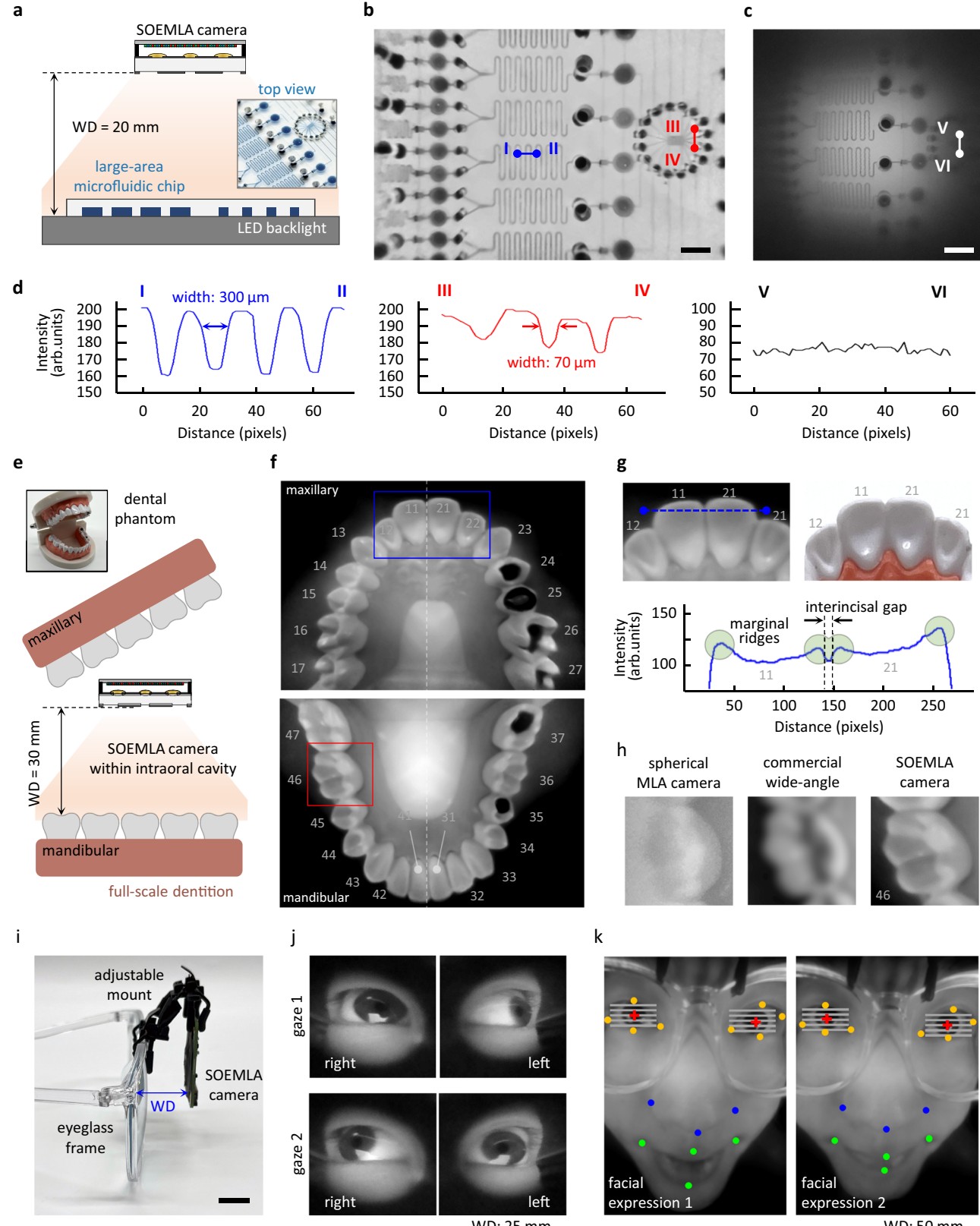

processing libraries, such as OpenCV, NumPy, and SciPy. The computational performance of the pipeline is benchmarked on a workstation equipped with an AMD Ryzen 9 7950X CPU (16 Cores) and 64 GB RAM, running Windows 11. No GPU acceleration is employed; instead, the processing is optimized using CPU-based parallel processing via Python's multiprocessing module. The processing time

for each stage is detailed in Supplementary Table 2. Further details on each algorithmic step are provided in the Methods section.

Stitching accuracy was quantitatively evaluated through image reprojection analysis. Reprojection accuracy was measured by comparing the ground-truth (GT) corner coordinates of ArUco markers with the reprojected points (RP) obtained from the estimated

**Fig. 5 | Exceptionally wide FOV imaging of real-world non-planar objects at short working distances using the SOEMLA camera. a** Experimental setup for capturing a large-area microfluidic chip (WD = 20 mm). Comparison of captured images between **b** the SOEMLA camera (FOV: 65 mm × 40 mm) and **c** a spherical MLA camera (FOV: 40 mm × 40 mm). Unlike the spherical MLA camera, the SOEMLA camera achieves uniform illumination and high contrast across the entire wide FOV. Scale bar: 5 mm. **d** Line intensity profiles corresponding to b and c. The reconstructed image from the SOEMLA camera clearly resolves microchannel structures with widths of 300 μm and 70 μm. **e** Experimental setup for capturing a full-scale adult oral phantom (WD = 30 mm). **f** Reconstructed images of maxillary and mandibular dentitions, labeled according to the ISO 3950 format. The SOEMLA camera allows for full-dentition imaging in a single acquisition within the confined oral cavity. **g** Enlarged views of the central incisors (teeth 11 and 21) alongside a commercial digital camera image. The intensity profile of the reconstructed image precisely reflects anatomical features such as the interincisal gap and marginal ridges. **h** Comparative images of a molar (tooth 46) using the spherical MLA camera, the commercial wide-angle camera, and the SOEMLA camera at a short WD. The SOEMLA camera demonstrates superior spatial resolution with clear structural boundaries. **i** Experimental setup for wearable facial imaging with an adjustable camera mount attached to an eyeglass frame. Scale bar: 20 mm. **j** Simultaneous imaging of both eyes under two distinct gaze directions at a WD of 25 mm. High-resolution capture at large angles provides reliable pupil identification for gaze tracking. **k** Reconstructed facial images at a WD of 50 mm under different expressions. The SOEMLA camera achieves effective full-face capture of dynamic facial features. Colored dots indicate facial landmarks.

homography matrix (Fig. 4b). Note that the same set of corner points was used for both reprojection analysis and image stitching. The reprojection errors remain low across the entire FOV with values of 1.05 pixels in the central (± 0–30°), 1.03 pixels in the mid-frame (± 15–55°), and 1.30 pixels in the peripheral region (± 30–70°) (Fig. 4c). The averaged reprojection error is 1.11 ± 0.20 pixels across all optical units, indicating consistently low errors despite substantial distortion at larger visual axis angles. Each partial image is spirally indexed from the center of 5 × 7 arrays (Supplementary Fig. 14). The low reprojection error relative to the fully stitched one-megapixel image reflects highly effective calibration and minimal aberration. Details of the pixel economy analysis are provided in Supplementary Note 2.

Experimental results confirm seamless wide FOV reconstruction by the SOEMLA camera. SOEMLAs captured partial images of insect targets (a wasp and a butterfly) displayed on an LCD screen at a WD of 20 mm (Fig. 4d). Each partial image corresponds to a distinct viewing angle, and adjacent images are overlapped by roughly 40% to provide redundant disparity for robust stitching. Final reconstructions show minimal aberrations and accurate scene reproduction over 120° horizontal, 90° vertical, and 140° diagonal FOV (Fig. 4e). Quantitative image quality was evaluated by comparing the reconstructed images with the ground truth using peak signal-to-noise ratio (PSNR) and structural similarity index (SSIM). In the experiment, the ground truth images were the original photographs displayed on the LCD. They were aligned to the reconstructed images via rotation and resizing to correct positional and resolution mismatches. The PSNR values are 21.46 dB for the wasp target and 18.72 dB for the butterfly target with corresponding SSIM values of 0.799 and 0.786. Magnified views of stitched boundaries confirm precise alignment and seamless edge integration, supporting the quantitative assessment (Fig. 4f). The SOEMLA camera's capability for video-rate capture and subsequent seamless wide FOV reconstruction of time-series data is further demonstrated in Supplementary Movie 1.

## Real-world demonstration of high-resolution wide field-of-view imaging

Multi-scale wide FOV imaging using the SOEMLA camera was conducted to capture real-world targets with complex surface geometries at short distances. A large-area microfluidic chip was captured under LED backlighting at 20 mm WD (Fig. 5a). The inset shows a poly-dimethylsiloxane (PDMS) microfluidic chip with blue-ink-filled microchannels of different widths. The SOEMLA camera captured a large FOV of 65 × 40 mm and produced uniform and high-contrast images across the entire area (Fig. 5b, Supplementary Fig. 15a). In contrast, the spherical MLA camera showed a 40 mm FOV with significant vignetting and peripheral contrast loss from optical aberrations using the same spherical MLA camera characterized in Fig. 3c. (Fig. 5c). The cross-sectional intensity profiles of microchannels clearly demonstrated the superior resolving power of the SOEMLA camera (Fig. 5d). For example, two representative regions (lines I–II and III–IV in Fig. 5b) clearly

resolved microfluidic features with widths of 300 μm and 70 μm, respectively. In comparison, the spherical MLA camera (line V–VI in Fig. 5c) produced low-modulation profiles and failed to resolve fine structures due to reduced resolution. The lens shading- and distortion-corrected result for the spherical MLA camera is shown in Supplementary Fig. 16. The SOEMLA camera provides high-resolution wide FOV imaging in a compact and short WD form for point-of-care applications.

A dental phantom was also captured to demonstrate the high-resolution wide FOV imaging within a confined space under realistic conditions. The SOEMLA camera was placed inside a full-scale dental phantom at 30 mm WD, reflecting the typical adult mouth opening range (Fig. 5e). Each of the maxillary and mandibular dentitions was captured in a single shot using the SOEMLA camera (Fig. 5f, Supplementary Fig. 15b). All 14 teeth in each dentition were precisely reconstructed and labeled according to ISO 3950 format, which confirms complete dental coverage. An enlarged view of the maxillary central incisors (teeth 11 and 21) demonstrated precise anatomical reconstruction (Fig. 5g). The interincisal gap and marginal ridges were clearly visible, and comparison with a reference commercial camera image validated visual consistency. Under identical conditions, the spherical MLA, commercial wide-angle, and SOEMLA cameras captured images, and resolution was compared using magnified views of posterior molar (tooth 46) (Fig. 5h). Unlike other cameras, the SOEMLA camera clearly resolved the tooth outline and internal textures owing to aberration reduction and wide FOV imaging at close range.

Wearable facial imaging was conducted to demonstrate the SOEMLA camera's wide FOV performance at close range. The SOEMLA camera was mounted on an eyeglass frame using an adjustable holder for head-mounted operation (Fig. 5i). The lightweight mount maintained stable positioning at close range and allowed adjustment of the camera angle and distance to the face. Both eyes were simultaneously imaged at 25 mm WD while the subject gazed in multiple directions (Fig. 5j). The contours of the pupil and iris were clearly visible even at large viewing angles, supporting the potential for accurate gaze estimation. Full-face imaging was also captured at a 50 mm WD while the subject displayed different facial expressions (Fig. 5k). Reconstructed images captured dynamic facial features such as changes in mouth shape, eye openness, and nose position. The experimental results confirm that the SOEMLA camera provides high-resolution and wide FOV imaging in a wearable format, maintaining image quality in the presence of facial motion and limited working space.

## Discussion
We have successfully demonstrated the SOEMLA camera for wide FOV imaging, inspired by the multi-angle chunk sampling strategy of *Xenos peckii*. The SOEMLA camera achieves 140° diagonal FOV and 0.94 mm TTL by integrating offset microapertures and ellipsoidal microlenses on a single CMOS ISA. Direction-specific spatial offsets and asymmetric microlens curvatures allow clear angular sampling with sharp PSFs at

oblique angles by effectively reducing optical aberrations. The wafer-level microfabrication provides precise control of ellipsoidal microlens curvature with an average error of 0.88%, yielding uniform resolution across the active pixel area of the CMOS ISA. The fully packaged SOEMLA camera significantly improves MTF50 compared to spherical MLA and commercial compact wide-angle cameras. Precise calibration enables seamless stitching of 5 × 7 partial images across a wide FOV, achieving total resolution of 1MP with 1.1-pixel reprojection error and average PSNR of 20.09 dB and SSIM of 0.793 values. Fully stitched images demonstrate high-resolution wide FOV imaging at shot distances for real-world targets such as a microfluidic chip, a dental phantom, and a human face. Unlike spherical MLAs and commercial wide-angle cameras, the SOEMLA camera captures fine features over large areas, independent of viewing angle and working distance. As summarized in Supplementary Table 3, this bioinspired engineering approach resolves the size-resolution trade-off in conventional wide FOV cameras, advancing space-constrained applications such as smart wearables and portable biomedical devices.

## Methods

### Microlens region-of-interest detection and elliptical masking

An automated masking procedure isolates individual microlens regions to extract reliable partial images from the raw SOEMLA camera data. A white reference image is first captured to enhance the visibility of partial image boundaries formed by each microlens. Pixels with intensity values above a global threshold T are selected. The selected bright pixel clusters are then grouped using morphological operations, and an ellipse ($\varepsilon_i$) is fitted using least-squares optimization for each group:

$$\varepsilon_i : Ax^2 + Bxy + Cy^2 + Dx + Ey + F = 0. \tag{1}$$

### Lens shading correction

Flat-field white images are acquired under uniform illumination to compensate for radial brightness fall-off due to oblique incidence and vignetting. A gain map $G(x, y)$ is computed by normalizing the flat-field response:

$$G(x, y) = \frac{I_{\text{ref}}}{I_{\text{flat}}(x, y)}. \tag{2}$$

### Distortion correction

Geometric distortions in each partial image are corrected based on the pinhole camera model described by Zhang[45]. Let $(x, y)$ be the distorted pixel coordinates and $(x_u, y_u)$ be the undistorted coordinates relative to the image center. The correction model simultaneously accounts for both radial and tangential distortions:

$$x_u = x\left(1 + k_1 r^2 + k_2 r^4 + k_3 r^6\right) + \left[2p_1 xy + p_2\left(r^2 + 2x^2\right)\right]$$

$$y_u = y\left(1 + k_1 r^2 + k_2 r^4 + k_3 r^6\right) + \left[p_1\left(r^2 + 2y^2\right) + 2p_2 xy\right] \tag{3}$$

where $r^2 = x^2 + y^2$. The five distortion parameters $(k_1, k_2, k_3, p_1, p_2)$ are estimated simultaneously with the camera's intrinsic matrix by optimizing the reprojection error from multiple views of the checkerboard calibration target using OpenCV's built-in functions.

### Homography estimation and registration

An ArUco marker array is used as the reference calibration pattern. Each microlens captures a distinct perspective of the planar target. For each microlens view, visible marker corners $\mathbf{x}_j$ are detected and matched with known marker positions $\mathbf{x}'_j$ in the global coordinate system.

A homography matrix $\mathbf{H}_i \in \mathbb{R}^{3 \times 3}$ is computed by solving:

$$\lambda \begin{bmatrix} x'_i \\ y'_i \\ 1 \end{bmatrix} = \mathbf{H}_i \begin{bmatrix} x_i \\ y_i \\ 1 \end{bmatrix}. \tag{4}$$

This transformation aligned each partial image to the global stitched frame. The reprojection error for each matched corner was calculated as

$$e_j = ||\mathbf{x}'_j - \mathbf{H}_i \mathbf{x_j}||_2. \tag{5}$$

### Poisson image blending

To create a seamless final image without visible edges from stitching, we employ Poisson image blending[46]. This technique edits an image $f$ (the final stitched canvas) by importing a source region $\Omega$ from another image $g$ (a warped partial image) while matching the gradients. The goal is to find an unknown function $f$ over $\Omega$ that minimizes the difference between its gradient field $\nabla f$ and the source gradient field $\mathbf{v} = \nabla g$. This is formulated as solving a Poisson equation with Dirichlet boundary conditions:

$$\min_f \iint_\Omega |\nabla f - \mathbf{v}|^2, \text{with} f|_{\partial\Omega} = g|_{\partial\Omega}. \tag{6}$$

This minimization is equivalent to solving the Poisson equation $\Delta f = \text{div}(\mathbf{v})$ over the domain $\Omega$, where $\Delta$ is the Laplacian operator. For the SOEMLA camera, the guidance field $\mathbf{v}$ is a composite of gradients from all overlapping warped images. The Dirichlet boundary conditions ($f|_{\partial\Omega}$) are defined by the pixel values at the outermost perimeter of the entire stitched canvas (i.e., regions covered by only one partial image). The resulting large, sparse linear system is formulated as a finite-difference problem and solved numerically using scipy.sparse.linalg.spsolve.

## Data availability

The authors declare that the main data supporting the findings of this study are available within the article and its Supplementary Information. In particular, representative raw images captured for camera calibration and wide field-of-view imaging are provided as Supplementary Data 1. Additional data that support the findings of this study are available from the corresponding authors upon request.

## Code availability

The authors declare that all image stitching and blending were implemented using established methods (Refs[45,46].) and built-in OpenCV functions, with no new algorithms developed and full reproducibility from the Methods. The computing code is available from the corresponding authors upon request.

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

## Acknowledgements

This work was supported by the National Research Foundation of Korea (NRF) grant funded by the Korean government (MSIT) (grant no. RS-2025-00523089 to J.M.K., Y.G.C., H.K.K., K.H.J.), the Korean ARPA-H Project through the Korea Health Industry Development Institute (KHIDI) funded by the Ministry of Health & Welfare (grant no. RS-2024-00512384 to J.M.K., Y.G.C., H.K.K., K.H.J.), and the Technology Innovation Program (grant no. RS-2024-00432381 to J.M.K., Y.G.C., H.K.K., K.H.J.) funded by the Ministry of Trade, Industry and Energy (MOTIE), Republic of Korea.

## Author contributions

Conceptualization: J.M.K., K.H.J., Y.G.C., H.K.K., M.H.K., J.K.P. Methodology: J.M.K., Y.K., Y.G.C., H.K.K. Investigation: J.M.K., Y.K., D.H.H. Visualization: J.M.K., Y.K., K.H.J., M.H.K. Funding acquisition: K.H.J. Project administration: K.H.J., J.M.K. Supervision: K.H.J., M.H.K., J.K.P. Writing—original draft: J.M.K., K.H.J. Writing – review & editing: J.M.K., K.H.J.

## Competing interests

The authors declare no competing interests.
