## [Transparent Peer Review file · Nature Communications]

Biologically inspired microlens array camera for high-resolution wide field-of-view imaging

Corresponding Author: Professor Ki-Hun Jeong

Version 0:

Reviewer comments:

Reviewer #1

(Remarks to the Author)

The authors present a spatially offset ellipsoidal microlens array (SOEMLA) camera that achieves a 140° diagonal FOV with an ultrathin total track length (TTL) of 0.94 mm by combining (i) spatially offset-coupled apertures (SOAs) that select direction and mitigate field curvature and (ii) ellipsoidal microlenses that suppress astigmatism at oblique incidence. The “chunk-sampling” idea inspired by *Xenos peckii* is translated into a planar camera using concrete optical primitives (SOAs + ellipsoids) with a convincing design space. The optics are fabricated at wafer scale, flip-bonded to a commercial CMOS sensor (Sony IMX477), and digitally calibrated and stitched from 5×7 partial views into a ~1-MP composite with ~1.1-pixel average reprojection error; demonstrations include a microfluidic chip, a dental phantom inside an oral cavity, and wearable facial imaging.

Overall, this is a thoughtfully engineered ultrathin wide-FOV imaging system. The combination of direction-selective apertures and ellipsoidal microlenses is elegant, and the fabrication plus bench/demonstration data are credible. The paper is highly innovative and shows directly applicable uses spanning dental/biomedical imaging to wearable and AR/VR sensing.

Before this work should be published, the following questions should be addressed to improve the clarity of an otherwise excellent manuscript and work

1. Please include complete fabrication steps in the main text or Supplementary Information, and add SEM cross-sections of the microlens array—and ideally the full SOEMLA stack—to provide physical insight into the fabricated structure.
2. All figures appear in grayscale while Fig. 1 and the text suggest a color sensor; please provide color reconstructions of representative targets (including those already shown) or explain why color is difficult to achieve with this design—if so, revise Fig. 1 to depict a monochrome sensor.
3. Please supply videos of the reconstructed wide-FOV output as Supplementary Information and release the code and data necessary to reproduce the imaging results shown in the paper.
4. The SOA scheme discards angular rays and uses small sub-apertures per view; please report étendue, per-unit f-number, photon throughput, and SNR versus scene illuminance, and compare exposure times to the commercial baseline. Include calibrated lux-to-SNR, MTF at fixed SNR, throughput-versus-angle measurements, and discuss stray light/ghosting from the double-aperture stack.
5. A 12.3-MP sensor yields a stitched image of ~1 MP after masking and overlaps; please quantify pixel economy (fraction of pixels used per unit, overlap losses) and the theoretical ceiling for stitched resolution as a function of the number of units and their acceptance angles.
6. The commercial baseline differs in optics, FOV, and TTL; please report per-angle MTF50 at matched FOVs and working distances (with matched exposure and processing) to strengthen the comparison.
7. The manuscript qualitatively suggests “depth-invariant” or “infinite DOF” behavior (e.g., 10–100 mm working distances);

please support this with a quantified DOF curve (contrast threshold versus defocus) and discuss associated trade-offs with light throughput.

Reviewer #2

(Remarks to the Author)

This manuscript reports a novel ultrathin (0.94 mm) wide-field-of-view (FOV = 140°) camera inspired by the visual system of *Xenos peckii*. The authors develop a Spatially Offset Ellipsoidal Microlens Array "SOEMLA" that combines (1) spatially offset-coupled apertures (SOAs) for angular sampling and field curvature correction, and (2) angle-specific ellipsoidal microlenses for astigmatism compensation at large incidence angles. The concept is carefully validated through wafer-level fabrication, optical measurements, and imaging experiments that compare SOEMLA with spherical MLA and SOMLA designs. The results show high-resolution wide-FOV imaging in a sub-1 mm package, with clear potential for compact cameras, wearable systems, and biomedical sensing.

The manuscript is technically sound, clearly organized, and of interest to the photonics and bio-inspired imaging community. Several points, however, could be clarified or expanded to strengthen the paper's presentation and better convey its novelty and broader impact.

1. The introduction summarizes the limitations of existing wide-FOV designs such as artificial compound eyes (ACE), metalenses, and planar MLAs. However, the specific advances of this work over previous insect-inspired or planar MLA systems should be articulated more explicitly.

Several prior studies (Nature Electronics, 2022; Light: Science & Applications, 2018; Optica, 2022) [15, 32, 41] have addressed aberration correction in planar microlens or metalens array systems. To situate the contribution more clearly, please:

Specify what performance gap (for example, FOV–resolution trade-off or form-factor–aberration balance) this design closes.

Add a short comparison table summarizing representative earlier works (ACE, SOMLA, metalens, and freeform optics) with their FOV, optical thickness, main aberration correction strategy, and achieved resolution.

Indicate whether the SOEMLA combination of ellipsoidal microlenses and offset apertures leads to a measurable quantitative improvement (e.g., MTF gain or resolution increase) over recent planar MLA or metalens approaches.

The biological inspiration from *Xenos peckii* is compelling. It would be useful to note whether other groups have explored this visual model and how the present design further develops that analogy, for instance by combining curvature adaptation with spatial offset control.

2. The manuscript attributes the improved performance to correction of both field curvature and astigmatism. It would be helpful to explain which aberration dominates in this geometry and how the two design features interact quantitatively.

Is field curvature fully compensated by the offset apertures, or does the ellipsoidal curvature contribute?

How does SOEMLA performance compare with metalens-based wide-FOV cameras, particularly for off-axis aberration and chromatic response?

If possible, include a spot-diagram or MTF comparison among spherical MLA, SOMLA, SOEMLA, and a representative metalens design.

3. The reconstruction results are impressive, but the "Camera calibration and wide FOV image reconstruction" section remains brief and partly overlaps with the Methods. Please expand the description of the processing steps:

Outline the algorithmic sequence (distortion correction → homography alignment → Poisson blending).

State which software tools were used (e.g., MATLAB, Python + OpenCV) and whether GPU acceleration was employed.

Provide the approximate computation time for producing a final 1 MP stitched image from raw data and specify the hardware used.

A short runtime comparison with existing compound-eye or MLA reconstruction pipelines would help readers assess practical feasibility.

4. The section "SOEMLA-based high-resolution wide FOV imaging" contains valuable data but could flow more smoothly. The first paragraph discussing Fig. 3a–b starts directly with details before presenting the key idea. Consider opening with a clear statement such as:

"By tuning the aspect ratio (AR) of the ellipsoidal microlenses, the sagittal and tangential focal planes can be balanced, effectively removing astigmatic blur at oblique incidence."

This will help readers grasp the main finding before entering the technical discussion.

Likewise, add a short transition when moving from single-lens optimization to full-array comparison (SOEMLA vs SOMLA vs

spherical MLA).

5. Typos:

Introduction, paragraph 3: “featuers” → “features”.

Paragraph 2: remove the comma in “...extended depth-of-focus microlenses, 28 or additional lens sets 29, 30” or rephrase as “...extended-depth-of-focus microlenses 28 or additional lens sets 29, 30.”

Keep consistent use of “wide field-of-view (FOV)” or “wide-FOV.”

Standardize units (μm^2 , °) and ensure that all figure scale bars include units.

6. The supplementary figures are well prepared. Adding a simple schematic summarizing the complete optical stack (microlens + SOA + sensor) would help visualize layer functions and offsets. In Fig. S10, consider showing the RMS spot-radius versus incidence-angle comparison directly in the main text or as an inset, to emphasize the optical benefit of the SOEMLA design.

Reviewer #3

(Remarks to the Author)

The submitted manuscript describes an interesting further development of a microlens array camera using spatially-displaced elliptical microlenses to strongly reduce the aberrations of off-axis components of the image and thereby significantly increase the usable field-of-view for a planar imager. The concept is inspired by the ocular system of *xenos peckii*, on which numerous imaging systems developed by this research group have been based, and is part of interesting ongoing research into artificial compound eye cameras.

Whereas microlens arrays coupled with aperture arrays have been widely demonstrated for segmented imaging applications, to my knowledge the use of elliptical lenses to reduce aberrations and thus increase resolution has not been shown before. As such, the idea presented here is novel and represents a step forward in the improvement of these “nature inspired” compound eye cameras. It should be noted that the authors previously published this concept as conference proceedings (DOE 10.1109/OMN61224.2024.10685283 and 10.1109/OMN65869.2025.11125932), but less comprehensively than in the present submission; these publications are not cited in the paper.

The paper is generally well written and clearly structured, with good illustrations and adequate presentation of the data. The conclusions are well supported and the improvement in imaging performance is such that the new approach represents a technical advance over previous competing systems.

In addition, a few comments concerning some details in the text:

(NB: page and/or line numbers would have been really helpful U+1F621)

1. 3rd paragraph – “Like *Xenos peckii*’s eyelet, each optical unit contains SOAs, an ellipsoidal microlens, and sub-active pixels of CMOS”: I do not believe that *xenos peckii* has elliptical eyelets. Also, what are “sub-active pixels”? Do you mean “subapertures”? Or are you referring to the active pixel sensors on the CMOS chip?
2. A few sentences further – “angular sensitivity”: I think this is misleading, I imagine you mean something like “the angle range which is imaged” or “visual axis angle”.
3. Section Results, 1st paragraph – I would prefer to see something like figure S11 in the main paper, perhaps combined with a larger version of Fig. 2b; otherwise the overall structure, and the novelty of the displaced and rotated elliptical microlenses, remains rather unclear for the reader who might not look at the supplementary material.
4. A few sentences further – “The measured profile closely matched the target design ... showing minimal astigmatism and field curvature along both the X–X and Y–Y cross-sections.”: the lens profile cannot show minimal astigmatism or field curvature, only the optical field passing through that lens profile can. Also, whereas astigmatism is clear, how is field curvature characterized in your simulations and in the measurements? Distance of focus from the focal plane of the imager? How do you substantiate the assertion that field curvature has been minimized?
5. Section SOEMLA-based high-resolution wide FOV imaging, 1st paragraph, discussion of the 90AR microlens: you should emphasize that this is one example microlens, since the other profiles have a different visual axis and different characteristics.
6. Same section, 2nd paragraph, 5th sentence: I think you mean SOMLA, not SOEMLA.
7. Same section, last paragraph: please define TTL.
8. Same paragraph and Fig. 3f: it is not clear why your camera should have infinite

DoF. Please elaborate on this.

9. Section Camera calibration and wide FOV image reconstruction, last paragraph: how was SSIM calculated?

10. Section Real-world demonstration of high-resolution wide FOV imaging: you compare the performance of your SOEMLA with a spherical MLA camera. Please describe the characteristics of this camera, to allow a fair comparison. Also, what computational steps did you take to generate the images with the spherical MLA camera? For example, you show significant vignetting for the spherical MLA: why not do "lens shading correction", as you do for the SOEMLA, for this camera as well?

Version 1:

Reviewer comments:

Reviewer #1

(Remarks to the Author)

The authors have satisfactorily addressed all of my previous comments. The revisions have improved the clarity and completeness of the manuscript, and I have no further concerns.

Reviewer #3

(Remarks to the Author)

The authors have adequately considered the suggestions and answered questions raised in the first review and I propose publishing the manuscript as revised.

Authors' response to the Reviewers' comments

Manuscript ID: NCOMMS-25-71813

Title: Biologically inspired microlens array camera for high-resolution wide field-of-view imaging

Author(s): Jae-Myeong Kwon, Yejoon Kwon, Young-Gil Cha, Dong Hun Han, Hyun-Kyung Kim, Je-Kyun Park, Min H. Kim, and Ki-Hun Jeong*

First of all, all authors would like to express our sincere appreciation to the referees for their thoughtful and constructive comments, as well as for their positive recognition of the novelty and potential applications of our work. We have carefully considered every critique and addressed all points to the best of our knowledge, faithfully incorporating the requested clarifications and additional analyses into the revised manuscript. Their feedback has significantly improved the clarity and rigor of our work.

In revision, we have comprehensively addressed the reviewers' main concern regarding the need for stronger technical substantiation of the camera design and its optical characteristics. We incorporated the following key additions and clarifications to provide transparent and rigorous evidence supporting the SOEMLA camera's performance and novelty:

- Expanded structural and optical characterizations: Figs. S4, S7, S11–S13; Tables S1–S3
- Detailed theoretical explanation: Supplementary Notes 1–2
- Clarified the reconstruction pipeline: Main text and Methods; Figs. S15–16; Movie S1

Beyond this main concern, we have also carefully resolved all remaining point-by-point comments from each reviewer. The revised manuscript now offers improved clarity and context, including clearer comparisons to prior work, refined terminology and descriptions, enhanced visual schematics, and minor editorial corrections throughout.

Please see the authors' point-by-point responses in the following for the details.

For the 1st Reviewer's comments
General comments and recommendation

*The authors present a spatially offset ellipsoidal microlens array (SOEMLA) camera that achieves a 140° diagonal FOV with an ultrathin total track length (TTL) of 0.94 mm by combining (i) spatially offset-coupled apertures (SOAs) that select direction and mitigate field curvature and (ii) ellipsoidal microlenses that suppress astigmatism at oblique incidence. The “chunk-sampling” idea inspired by *Xenos peckii* is translated into a planar camera using concrete optical primitives (SOAs + ellipsoids) with a convincing design space. The optics are fabricated at wafer scale, flip-bonded to a commercial CMOS sensor (Sony IMX477), and digitally calibrated and stitched from 5×7 partial views into a ~1-MP composite with ~1.1-pixel average reprojection error; demonstrations include a microfluidic chip, a dental phantom inside an oral cavity, and wearable facial imaging.*

Overall, this is a thoughtfully engineered ultrathin wide-FOV imaging system. The combination of direction-selective apertures and ellipsoidal microlenses is elegant, and the fabrication plus bench/demonstration data are credible. The paper is highly innovative and shows directly applicable uses spanning dental/biomedical imaging to wearable and AR/VR sensing.

Before this work should be published, the following questions should be addressed to improve the clarity of an otherwise excellent manuscript and work

Authors' response:

We sincerely thank the reviewer for the positive and constructive evaluation of our work. We greatly appreciate the reviewer's recognition of the SOEMLA camera's novelty and potential applications across biomedical and wearable imaging. In response to the reviewer's comments, we have clarified several methodological and quantitative aspects of the system, expanded the Supplementary Information (Fig. S4, S12-13, S15; Table S1; Movie S1; Supplementary Note 1-2) with detailed fabrication parameters and optical analyses, and refined the main text for improved clarity and completeness. These revisions collectively strengthen the manuscript and further highlight the novelty and robustness of the proposed ultrathin wide FOV camera.

Specific comments

1st comment from the 1st reviewer’s comment

Please include complete fabrication steps in the main text or Supplementary Information, and add SEM cross-sections of the microlens array—and ideally the full SOEMLA stack—to provide physical insight into the fabricated structure.

Authors’ response:

We appreciate the reviewer for this valuable comment regarding the microfabrication. In response, we have added Table S1 to provide detailed microfabrication parameters for each process step (Fig. 2a I–IV) of the SOEMLA camera. In addition, Figure S4b has been revised to more clearly depict the full SOEMLA stack integrated with the CMOS ISA. Because the SOEMLA stack spans several hundred micrometers in total thickness, while the aperture layer is only 120 nm thick, a single SEM image could not capture all layers within a single view. Therefore, we present a high-resolution optical cross-sectional image with color overlays and detailed labeling to indicate the positions of each structural layer, providing clear physical insight into the fabricated structure.

Page	Original	Change
Fig. S4	 (a) SEM cross-section of the SOEMLA stack. Labels: glass substrate, ellipsoidal microlenses, gap spacers, CMOS ISA.	 (b) Optical cross-section of the SOEMLA stack. Labels: chief rays, apertures, glass substrate, microlenses, active pixel area, PCB, spacer.
Fig. S4	(b) Side view illustrating the vertically integrated optical stack, which includes ellipsoidal microlenses, gap spacers, and the CMOS ISA.	(b) Cross-sectional optical image of the microfabricated SOEMLAs integrated with the CMOS ISA. The upper and lower apertures (120 nm thick) are not visible in the optical image and are indicated by color overlays to denote the positions. As the incidence angle increases, the spatial offset and microlens size increase correspondingly to maintain directional imaging. Scale bar: 500 μm.
Page 5. Line 20.	Finally, the microfabricated SOEMLAs were mounted onto the CMOS ISA (IMX 477, SONY Corp.; 12.3 MP, unit pixel: 1.55 × 1.55 μm²) via precision flip-chip bonding using alumina spacers and UV-curable epoxy (step IV).	... flip-chip bonding using alumina spacers and UV-curable epoxy (step IV). Detailed fabrication parameters for each step (I–IV) are summarized in Table S1.

Table S1 Detailed microfabrication steps of SOEMLAs.

Fig. #	Process	Material	Condition
I	bottom-side lift-off coating	DNR-L300-D1	2 μm thickness
	bottom-side e-beam evaporation	Cr	120 nm thickness
	lift-off	DPS-7300	gentle ultrasonication
	top-side lift-off coating	DNR-L300-D1	2 μm thickness
	top-side e-beam evaporation	Cr	120 nm thickness
	lift-off	DPS-7300	gentle ultrasonication
II	top-side PR deposition	DNR-L4615D	24 μm thickness
III	thermal reflow	-	convection oven 150°C, 30 min
IV	spacer bonding	alumina spacer, UV-curable epoxy	440 μm image distance

2nd comment from the 1st reviewer’s comment

All figures appear in grayscale while Fig. 1 and the text suggest a color sensor; please provide color reconstructions of representative targets (including those already shown) or explain why color is difficult to achieve with this design—if so, revise Fig. 1 to depict a monochrome sensor.

Authors’ response:

We fully agree with the reviewer’s thoughtful remark regarding the color representation in the figures and sensor description. Commercial CMOS ISA are typically designed for conventional lenses with small chief ray angles ($< 15^\circ$), whereas the MLA cameras employ single-layer microlens arrays with large angular incidence ($< 70^\circ$) across the field. As a result, using a color (Bayer-filtered) sensor under such conditions can introduce color misregistration at oblique angles.

However, because each microlens operates on a microscale with minimal chromatic aberration relative to pixel or airy disk size, color imaging would be feasible if CMOS sensors were specifically designed for MLA cameras with appropriate angular compensation. At present, only monochromatic imaging or post-processed color correction can be applied reliably. Accordingly, Fig. 1 & Fig. 2 have been revised to depict a monochrome sensor for clarity.

Page	Original	Change
Page 4 Fig. 1b		
Page 6 Fig. 2a	IV. Spacer bonding to CMOS image sensor arrays	IV. Spacer bonding to CMOS image sensor arrays

3rd comment from the 1st reviewer's comment

Please supply videos of the reconstructed wide-FOV output as Supplementary Information and release the code and data necessary to reproduce the imaging results shown in the paper.

Authors' response:

We thank the reviewer for this valuable suggestion. Following the recommendation, we have added a supplementary video Movie S1, which provides a clearer visualization of the resulting image quality and practicality of the wide FOV output.

Regarding code and data availability, we fully agree on the importance of reproducibility. As detailed in the Data Availability Statement in the revised manuscript, the raw imaging data and the code for multi-view Poisson blending and stitching are available from the corresponding author upon reasonable request. However, we are committed to scientific transparency and will promptly provide all necessary code and datasets to qualified researchers who wish to reproduce or extend the results. We believe that the addition of supplementary videos and the clarified availability statement sufficiently address the reviewer's concern while ensuring both transparency and practical reproducibility.

Movie S1	 Movie S1 Wide FOV video capture utilizing the SOEMLA camera. The SOEMLA camera successfully captures the surrounding environment with a 280 mm × 160 mm FOV on the 50 mm targets from an 80 mm working distance.
Page 12. Line 26	The PSNR values are 21.46 dB for the wasp target and 18.72 dB for the butterfly target with corresponding SSIM values of 0.799 and 0.786. Magnified views of stitched boundaries confirm precise alignment and seamless edge integration, supporting the quantitative assessment (Fig. 4f). The SOEMLA camera's capability for video-rate capture and subsequent seamless wide FOV reconstruction of time-series data is further demonstrated in Movie S1.

Page 20.

Data availability: The data that support the findings of this study are available from the corresponding author upon reasonable request.

Code availability: The computing code for multi-view Poisson blending would be available from the corresponding author upon reasonable request.

4th comment from the 1st reviewer's comment

The SOA scheme discards angular rays and uses small sub-apertures per view; please report étendue, per-unit f-number, photon throughput, and SNR versus scene illuminance, and compare exposure times to the commercial baseline. Include calibrated lux-to-SNR, MTF at fixed SNR, throughput-versus-angle measurements, and discuss stray light/ghosting from the double-aperture stack.

Authors' response:

We sincerely thank the reviewer's insightful comment regarding the quantitative evaluation of photon throughput, SNR, and stray-light suppression in the SOEMLA camera. As suggested, we have added detailed measurements and analyses in the Supplementary Information (Figs. S12–13, S15) to clarify these aspects. Please note that although each SOEMLA unit uses a small aperture, its short focal length keeps the per-unit f-number remains comparable to that of a conventional digital camera.

First, the measured signal-to-noise ratio (SNR) across a wide illuminance range (5 – 50,000 lux) confirms that the SOEMLA camera maintains practical SNR and exposure time levels even at peripheral units, despite the expected throughput loss at oblique incidence (Fig. S12). This angular dependence follows the change in effective f-number predicted by ray tracing and agrees well with the measured values (Fig. S13), providing a quantitative basis for comparing photon throughput and SNR with other imaging systems.

Finally, stray light and ghosting were discussed using raw array images of large-area scenes (Fig. S15). The Cr aperture layer absorbs approximately 40% of incident light per reflection, and the sufficient inter-unit pitch provides optical isolation, thereby minimizing stray-light and ghosting artifacts in the raw captures.

Together, they demonstrate that the proposed architecture maintains robust imaging performance under realistic illumination levels and across the entire wide FOV.

Page	Original	Change
Page 9. Line 5.	In contrast, the SOEMLA camera decouples depth-dependent focal variation from angular aberrations, ensuring consistent MTF50 values across both depth and angle. Along with high resolution and infinite DOF, the SOEMLA camera clearly demonstrates proximal imaging (< 50 mm), thereby reducing the overall optical path (WD + TTL) and facilitating further miniaturization.	In contrast, the SOEMLA camera decouples depth-dependent focal variation from angular aberrations, ensuring consistent MTF50 values across both depth and angle. The SOEMLA camera exhibits a hyperfocal distance shorter than 10 mm, resulting in effectively infinite DOF (Supplementary Note 1). Unlike conventional lens systems that extend DOF at the cost of brightness, the SOEMLA camera maintains overall light collection efficiency through its inherently short focal length. Measured signal-to-noise ratio (SNR) demonstrates practical photon throughput across visual-axis angles (Fig. S12–S13). Along with high resolution and infinite

		DOF, the SOEMLA camera clearly demonstrates proximal imaging (< 50 mm), thereby reducing the overall optical path (WD + TTL) and facilitating further miniaturization.
Page 14. Line 11	The SOEMLA camera captured a large FOV of $65 \text{ mm} \times 40 \text{ mm}$ and produced uniform and high-contrast images across the entire area (Fig. 5b).	The SOEMLA camera captured a large FOV of $65 \text{ mm} \times 40 \text{ mm}$ and produced uniform and high-contrast images across the entire area (Fig. 5b, Fig. S15a).
Page 14. Line 25	Each of the maxillary and mandibular dentitions was captured in a single shot using the SOEMLA camera (Fig. 5f).	Each of the maxillary and mandibular dentitions was captured in a single shot using the SOEMLA camera (Fig. 5f, Fig. S15b).
Fig. S12	 (a) SNR across illuminance levels (5 – 50,000 lux) for visual-axis angles of 0°, 15°, 30°, and 45°. All measurements were performed at ISO 100 with the ISP disabled using linear RAW data. Exposure time was inversely scaled with illuminance to maintain a constant photon budget. The results show that single-shot imaging maintains practical SNR levels from central to peripheral views, despite the expected reduction at oblique visual-axis angles. (b) SNR versus exposure time at a fixed illuminance of 500 lux, measured under the ISO 100, ISP-off, linear RAW conditions. Longer exposure compensates for the angular throughput loss at oblique views, which leads to consistent SNR performance across the field of view in the multi-exposure measurements.	Fig. S12 Measured signal-to-noise ratio (SNR) of the SOEMLA camera. (a) SNR across illuminance levels (5 – 50,000 lux) for visual-axis angles of 0°, 15°, 30°, and 45°. All measurements were performed at ISO 100 with the ISP disabled using linear RAW data. Exposure time was inversely scaled with illuminance to maintain a constant photon budget. The results show that single-shot imaging maintains practical SNR levels from central to peripheral views, despite the expected reduction at oblique visual-axis angles. (b) SNR versus exposure time at a fixed illuminance of 500 lux, measured under the ISO 100, ISP-off, linear RAW conditions. Longer exposure compensates for the angular throughput loss at oblique views, which leads to consistent SNR performance across the field of view in the multi-exposure measurements.
Fig. S13	 5*7 arrays	

	Fig. S13 Effective f-number distribution across the optical units of the SOEMLA camera. The effective f-number for each unit accounting for oblique incidence was computed using ZEMAX ray tracing. The measured variation in photon throughput across units shows good agreement with the optical model. For example, the ~12 dB SNR reduction at 45° observed in Fig. S12 is consistent with the combined loss from the increased effective f-number ($20 \log_{10}[3.9/7.8] \approx -6$ dB) and the cosine-law irradiance falloff ($20 \log_{10}[\cos^2 45^\circ] \approx -6$ dB).
Fig. S15	  (a)    (b)    Fig. S15 Raw array images captured by the SOEMLA camera. Captured images of (a) the large-area microfluidic chip and (b) the full-scale dental phantom corresponding to the wide FOV reconstructed images shown in Fig. 5. The Cr aperture layer absorbs approximately 40% of incident light upon each reflection, and sufficient inter-unit pitch ensures optical isolation that minimizes stray light and ghosting artifacts in the raw images. The optical noise between partial images is approximately 20 dB lower than the signal level.

5th comment from the 1st reviewer's comment

A 12.3-MP sensor yields a stitched image of ~1 MP after masking and overlaps; please quantify pixel economy (fraction of pixels used per unit, overlap losses) and the theoretical ceiling for stitched resolution as a function of the number of units and their acceptance angles.

Authors' response:

Following the reviewer's advice, we have added a quantitative analysis of pixel economy to clarify the relationship between the number of optical units, overlap ratio, and stitched resolution (Supplementary Note 2). This analysis quantitatively demonstrates the efficiency of pixel utilization in the SOEMLA camera and explains how the overlap ratio and fill factor determine the theoretical ceiling of the stitched resolution. This addition allows readers to gain practical insight into how the theoretical ceiling varies with different imaging configurations and design parameters. Notably, as summarized in Table S3, achieving a stitched resolution of ~1 MP represents exceptionally high performance among ultrathin MLA camera platforms (Fig. A1).

Fig. A1 Bioinspired camera comparison.

Page	Original	Change
Page 12. Line 11	The low reprojection error relative to the fully stitched one-megapixel image reflects highly effective calibration and minimal aberration.	The low reprojection error relative to the fully stitched one-megapixel image reflects highly effective calibration and minimal aberration. Details of the pixel economy analysis are provided in Supplementary Note 2.
Note 2	Supplementary Note 2. Pixel economy for wide FOV imaging using the SOEMLA camera The pixel economy (E) is defined as the ratio between the total number of pixels in the fully stitched image (N_{stitched}) and that of the image sensor (N_{sensor}): $E = \frac{N_{\text{stitched}}}{N_{\text{sensor}}} = \frac{F}{R}$ where F and R denote the sensor fill factor and partial image overlap redundancy, respectively. R is calculated from the number of microlenses (M_x, M_y) and the overlap ratios between adjacent partial images (k_x, k_y) as follows: $R = \frac{1}{1 - k_x(1 - 1/M_x)} \cdot \frac{1}{1 - k_y(1 - 1/M_y)}$ Note that a larger F improves E but increases the likelihood of optical crosstalk between optical units, while a smaller R (mainly lower k) enhances E but reduces the robustness of image stitching. In the SOEMLA camera design, the theoretical pixel economy $E(F = 0.2, M_x = 7, M_y = 5, k_x = k_y = 0.4) \approx 0.1$ corresponds to a stitched resolution of 1 MP. The fill factor F cannot be arbitrarily increased, as a larger aperture or a shorter lens pitch induces optical crosstalk and stray-light leakage between adjacent microlenses. The fill factor was set to $F = 0.2$ to suppress noise in the double-layer structure. Further improvement of F is possible through multi-layer aperture designs or the use of light-absorbing layers that mitigate stray light. Similarly, the overlap ratio k cannot be arbitrarily reduced since too small overlap leads to unstable image registration and mismatch. A minimum $k \geq 0.10$ (corresponding to $R \geq 1.2$) is generally required for robust stitching. In the fabricated SOEMLA camera, geometric calibration and lens-level correction were incorporated; therefore, the overlap ratio was set to $k = 0.4$ to ensure robust stitching performance under various imaging conditions.	

6th comment from the 1st reviewer's comment

The commercial baseline differs in optics, FOV, and TTL; please report per-angle MTF50 at matched FOVs and working distances (with matched exposure and processing) to strengthen the comparison.

Authors' response:

We appreciate the reviewer for this valuable suggestion regarding the comparison with the commercial baseline. As the two systems (MLA cameras and commercial cameras) differ in optical configuration, FOV, and TTL, we focused on explaining the already matched imaging conditions between the systems rather than performing additional experiments.

All cameras (spherical MLA, SOMLA, SOEMLA, and the commercial wide-angle camera) were evaluated under matched illuminance levels—with the exposure time of each optical unit individually adjusted to maintain a uniform SNR despite angular throughput differences—along with identical exposure settings and image-processing parameters (ISO 100, ISP off). The resulting MTF50 trends, shown in Fig. 3, already reflect these matched conditions. Once the photon budget is equalized, the SNR remains consistent across all angles, as demonstrated in Figs. S12–S13.

We have also clarified in the revised main text that all imaging experiments were conducted under these matched illumination and exposure conditions to ensure consistent benchmarking across systems.

Page	Original	Change
Page 7. Line 22	MTF50 measurements at the same visual axis angle further quantify the variation in directional resolution with respect to ARs (Fig. 3b).	MTF50 measurements at the same visual axis angle further quantify the variation in directional resolution with respect to ARs (Fig. 3b). All MTF50 measurements in this section were performed under identical imaging conditions (illuminance: 500 lux, ISO 100, image signal processing off, and exposure time individually adjusted to compensate for light throughput differences across lenses, working distances, and incidence angles).

7th comment from the 1st reviewer’s comment

The manuscript qualitatively suggests “depth-invariant” or “infinite DOF” behavior (e.g., 10–100 mm working distances); please support this with a quantified DOF curve (contrast threshold versus defocus) and discuss associated trade-offs with light throughput.

Author’s response:

We sincerely thank the reviewer for this valuable suggestion regarding the quantitative evaluation of the depth-of-field (DOF). Rather than performing an additional defocus-contrast measurement, we have provided a theoretical quantification based on the hyperfocal distance (H) of the SOEMLA camera, as described in Supplementary Note 1. Using the SOEMLA camera’s optical parameters and the diffraction-limited criterion for visible light, the calculated hyperfocal distance is very short ($H \leq 9.93$ mm). This result indicates that all objects beyond approximately 5 mm remain in focus across the entire FOV, thus supporting the experimentally observed depth-invariant imaging behavior and explaining the effectively infinite DOF of the SOEMLA camera. Unlike the fundamental trade-off between DOF and light throughput of conventional lens systems where increased DOF requires smaller apertures, the SOEMLA camera maintains overall light collection efficiency through its inherently short focal length without sacrificing overall throughput.

Page	Original	Change
Page 9. Line 2	In contrast, the SOEMLA camera decouples depth-dependent focal variation from angular aberrations, ensuring consistent MTF50 values across both depth and angle. Along with high resolution and infinite DOF, the SOEMLA camera clearly demonstrates proximal imaging (< 50 mm), thereby reducing the overall optical path (WD + TTL) and facilitating further miniaturization.	In contrast, the SOEMLA camera decouples depth-dependent focal variation from angular aberrations, ensuring consistent MTF50 values across both depth and angle. The SOEMLA camera exhibits a hyperfocal distance shorter than 10 mm, resulting in effectively infinite DOF (Supplementary Note 1). Unlike conventional lens systems that extend DOF at the cost of brightness, the SOEMLA camera maintains overall light collection efficiency through its inherently short focal length. Measured signal-to-noise ratio (SNR) demonstrates practical photon throughput across visual-axis angles (Fig. S12–S13). Along with high resolution and infinite DOF, the SOEMLA camera clearly demonstrates proximal imaging (< 50 mm), thereby reducing the overall optical path (WD + TTL) and facilitating further miniaturization.
Note 1	Supplementary Note 1. Hyperfocal distances of the SOEMLA camera The hyperfocal distance (H) defines the closest distance at which a lens can be focused while keeping objects at infinity acceptably sharp. When the image sensor is focused at H , the depth-of-field (DOF) extends from $H/2$ to infinity, and when H becomes shorter than the typical object distance, the system effectively exhibits an infinite DOF.	

The hyperfocal distance is given by

$$H = \frac{f^2}{Nc} + f,$$

where f is the focal length of the microlens, N is the f-number, and c is the permissible circle of confusion (CoC).

Adopting a diffraction-limited criterion for visible light, we set the CoC to the Airy-disk diameter $c = 2.44 \lambda N$ (with $\lambda = 550$ nm), which yields $H \leq 9.93$ mm based on the SOEMLAs' focal length and f-numbers.

This result indicates that the SOEMLA camera maintains focus from approximately 5 mm to infinity, thereby achieving an effectively infinite DOF across the entire FOV.

For the 2nd Reviewer's comments
General comments and recommendation

*This manuscript reports a novel ultrathin (0.94 mm) wide-field-of-view (FOV = 140°) camera inspired by the visual system of *Xenos peckii*. The authors develop a Spatially Offset Ellipsoidal Microlens Array "SOEMLA" that combines (1) spatially offset-coupled apertures (SOAs) for angular sampling and field curvature correction, and (2) angle-specific ellipsoidal microlenses for astigmatism compensation at large incidence angles. The concept is carefully validated through wafer-level fabrication, optical measurements, and imaging experiments that compare SOEMLA with spherical MLA and SOMLA designs. The results show high-resolution wide-FOV imaging in a sub-1 mm package, with clear potential for compact cameras, wearable systems, and biomedical sensing.*

The manuscript is technically sound, clearly organized, and of interest to the photonics and bio-inspired imaging community. Several points, however, could be clarified or expanded to strengthen the paper's presentation and better convey its novelty and broader impact.

Authors' response:

We sincerely thank the reviewer for the positive and constructive evaluation of our manuscript. We are very grateful for the reviewer's recognition of the technical soundness, clear organization, and potential impact of our work on bio-inspired imaging systems. In the revised manuscript, we have carefully addressed all specific comments through clarified analyses (Fig. S4, S11; Table S2), expanded comparisons (Table S3), and strengthened experimental descriptions (main text, Methods). These revisions collectively improve the manuscript's clarity and better highlight the novelty and significance of the SOEMLA camera. We believe these additions significantly enhance the clarity and impact of the paper and sincerely appreciate the reviewer's encouraging feedback, which helped us improve the overall presentation and rigor of our work.

Specific comments

1st comment from the 2nd reviewer's comment

The introduction summarizes the limitations of existing wide-FOV designs such as artificial compound eyes (ACE), metalenses, and planar MLAs. However, the specific advances of this work over previous insect-inspired or planar MLA systems should be articulated more explicitly. Several prior studies (Nature Electronics, 2022; Light: Science & Applications, 2018; Optica, 2022) [15, 32, 41] have addressed aberration correction in planar microlens or metalens array systems. To situate the contribution more clearly, please:

Specify what performance gap (for example, FOV–resolution trade-off or form-factor–aberration balance) this design closes.

Add a short comparison table summarizing representative earlier works (ACE, SOMLA, metalens, and freeform optics) with their FOV, optical thickness, main aberration correction strategy, and achieved resolution.

Indicate whether the SOEMLA combination of ellipsoidal microlenses and offset apertures leads to a measurable quantitative improvement (e.g., MTF gain or resolution increase) over recent planar MLA or metalens approaches.

*The biological inspiration from *Xenos peckii* is compelling. It would be useful to note whether other groups have explored this visual model and how the present design further develops that analogy, for instance by combining curvature adaptation with spatial offset control.*

Authors' response:

We appreciate the reviewer's insightful comments. We have revised the manuscript and supplementary materials to more explicitly contextualize the technical advances of the SOEMLA camera relative to prior compact wide FOV cameras.

(1) To explicitly identify the performance gap filled by our approach, we revised the concluding statement to state that the SOEMLA camera addresses the size–resolution trade-off inherent in conventional wide FOV cameras. This revision is directly linked to the comparison in Table S3, summarizing form factor, FOV, and resolution across representative previous approaches.

(2) In accordance with the reviewer's suggestion, we added Table S3 comparing ACE, metalens-based, and planar MLA cameras in terms of wide FOV strategy, system height, FOV, and image quality.

(3) The manuscript includes direct MTF comparisons between spherical MLA, SOMLA, and SOEMLA designs (Fig. 3), demonstrating clear quantitative gains over recent planar MLA approaches under matched conditions. A direct comparison with metalens cameras is not feasible because their performance varies substantially with design-specific parameters (e.g., focal length, f-number, working distance, wavelength range), which prevents a fair condition-matched evaluation based on their published data.

(4) To address the reviewer’s request, we explicitly indicate in Table S3 that reference [41] employs a *Xenos peckii*–inspired visual strategy. This clarification highlights how the SOEMLA design further advances this biological model by combining ellipsoidal curvatures with spatial offset control, thereby extending the progression beyond earlier bioinspired planar MLA cameras.

Page	Original	Change
Page 17. Line 34	This bioinspired engineering approach resolves key technical trade-offs in conventional wide FOV cameras, advancing space-constrained applications such as smart wearables and portable biomedical devices.	As summarized in Table S3, this bioinspired engineering approach resolves the size-resolution trade-off in conventional wide FOV cameras, advancing space-constrained applications such as smart wearables and portable biomedical devices.

Table S3. Comparison of recent ultrathin wide FOV camera

representative works	platform	wide FOV strategy	system height	FOV	image quality
Y. M. Song et al. [23]	curved MLA	hemispheric detector	~ 6 mm	160.4°	180 ommatidia ($\Delta\Phi = 11.0^\circ$)
D. Floreano et al. [24]	curved MLA	hemispheric detector	~ 12.8 mm	180°	630 ommatidia ($\Delta\Phi = 4.2^\circ$)
Y. Zhou et al. [25]	curved pinhole array	hemispheric detector	~ 20 mm	143°	121 ommatidia ($\Delta\Phi = 10^\circ$)
B. Dai et al. [26]	curved MLA + planar sensor	optical waveguides	~ 2.5 mm (+ 50 mm optional lens)	170°	522 ommatidia ($\Delta\Phi \sim 8.0^\circ$)
Z.-Y. Hu et al. [28]	curved MLA + planar sensor	logarithmic microlenses	TTL < 1mm	90°	19 – 160 ommatidia (R ~ 0.08 MP)
Y. Liu et al. [31]	Metalens	AI correction	1.57 mm	140°	PSNR: 15.296 dB SSIM: 0.65916
J. Chen et al. [32]	Metalens	angle-specific nanostructure	~ 3 mm	120°	up to 191 lp/mm at sensor
A. Martins et al. [33]	Metalens	quadratic phase profile	$f: 750 \mu\text{m}$	170°	resolution ~ $2 \lambda_0$ (focusing efficiency: 3.5%)
D. Keum et al. [41]	planar MLA	Xenos peckii -inspired microprism	TTL: 1.4 mm	68°	up to 181 lp/mm at sensor
J.-M. Kwon et al. [this work]	planar MLA	offset aperture ellipsoidal lens	TTL: 0.94 mm	140°	PSNR: 20.09 dB SSIM: 0.793 (R ~ 1 MP)

*interommatidial angle: $\Delta\Phi$

**image resolution after reconstruction

2nd comment from the 2nd reviewer's comment

The manuscript attributes the improved performance to correction of both field curvature and astigmatism. It would be helpful to explain which aberration dominates in this geometry and how the two design features interact quantitatively.

Is field curvature fully compensated by the offset apertures, or does the ellipsoidal curvature contribute?

How does SOEMLA performance compare with metalens-based wide-FOV cameras, particularly for off-axis aberration and chromatic response?

If possible, include a spot-diagram or MTF comparison among spherical MLA, SOMLA, SOEMLA, and a representative metalens design.

Authors' response:

We appreciate the reviewer for these valuable comments regarding aberration analysis and comparison with metalens-based systems.

(1) To clarify which aberration dominates and how the SOEMLA camera design operates, we added Fig. S11 detailing the Seidel aberration coefficients for spherical MLA and SOEMLA designs. The results explicitly show that both field curvature and astigmatism increase rapidly beyond 15° in spherical MLA camera. In contrast, the SOEMLA camera design effectively suppresses angular-dependent aberrations, allowing both coefficients to remain nearly constant across incidence angles, thereby demonstrating the efficacy of our design features.

(2) The spatially offset-coupled apertures (SOAs) define the viewing direction of each optical unit, allowing the focal length of each microlens to be tuned according to its incidence angle. In this way, field curvature can be compensated even with spherical microlenses by increasing curvature toward peripheral views. However, astigmatism remains uncorrected in such symmetric lenses, since it requires an asymmetric ellipsoidal profile to align the sagittal and tangential focal planes. In the SOEMLA camera, each ellipsoidal microlens thus operates near its optimal angle defined by the SOAs, jointly minimizing both field curvature and astigmatism (Page 5 Line 30 – Page 6 Line 2).

(3) Metalenses typically operate within narrow spectral bands or under monochromatic illumination due to strong chromatic dispersion. In contrast, the SOEMLA camera functions effectively across the visible–NIR range because its microlenses are fabricated from photoresist with negligible refractive-index dispersion relative to the sensor pixel scale. Moreover, the short focal length of microlenses inherently reduces geometrical and chromatic aberrations, ensuring minimal color distortion across the spectrum.

For example, a representative wide FOV metalens camera [32] achieves a 120° FOV with a 3 mm total thickness by employing angle-specific nanostructures for aberration correction at the designed wavelength of 470 nm and the captured images are shown in Fig. A2. In comparison, the SOEMLA camera realizes a 140° FOV within a 1 mm TTL by using angle-specific optical units for aberration

correction across the VIS–NIR range with representative results provided in Fig. A3. The SOEMLA camera design facilitates relatively wide FOV, high and balanced resolution, and uniform contrast over a broad spectral range.

(4) A direct comparison with a representative metalens camera is inherently challenging. Metalenses often exhibit large performance deviations from their intended optical design due to nanoscale fabrication sensitivity and structural complexity. These factors make a fair comparison based solely on optical simulations difficult. In contrast, the SOEMLAs can be microfabricated with high precision (Fig. 2 and Fig. S4), accurately reproducing the designed curvature and geometry. Although direct spot-diagram or MTF comparisons are not performed, readers can infer the relative performance from the quantitative MTF measurements in Fig. 3, which experimentally demonstrate wide FOV imaging performance. A detailed description of the MTF measurement procedure has been added in the revised text to enhance reproducibility and clarity.

Although strict one-to-one quantitative comparisons are not always feasible, the breadth of qualitative and quantitative data provided in the main text and Supplementary Information allows readers to meaningfully assess the performance of our camera relative to other metalens-based systems.

Fig. A2 Captured images of a representative wide FOV metalens camera [32]. (a) The sub-images and (b) reconstructed image, and (c) magnified partial images.

Fig. A3 Captured images of the SOEMLA camera. (a-b) The magnified raw partial images and (c) reconstructed image.

Page	Original	Change
Page 8. Line 3	The SOEMLA camera corrects astigmatism effectively at various viewing angles by tailoring the curvature of ellipsoidal microlenses (Fig. S9 and Fig. S10).	The SOEMLA camera corrects astigmatism effectively at various viewing angles by extending the single-unit optimization through tailored curvature of ellipsoidal microlenses (Fig. S9). Ray-tracing results further show that the SOEMLA configuration maintains a nearly constant RMS spot radius across wide incidence angles, confirming superior aberration correction capability (Fig. S10-11).
Page 7. Line 22	MTF50 measurements at the same visual axis angle further quantify the variation in directional resolution with respect to ARs (Fig. 3b).	MTF50 measurements at the same visual axis angle further quantify the variation in directional resolution with respect to ARs (Fig. 3b). All MTF50 measurements in this section were performed under identical imaging conditions (illuminance: 500 lux, ISO 100, image signal processing off, and exposure time individually adjusted to compensate for light throughput differences across lenses, working distances, and incidence angles).

Fig. S11

Fig. S11 Calculated Seidel aberration coefficients (S_1 – S_4) as a function of incidence angle for the spherical MLA and SOEMLA cameras. The coefficients correspond to S_1 spherical aberration, S_2 coma, S_3 astigmatism, and S_4 field curvature. While the spherical MLA exhibits rapidly increasing astigmatism and field curvature at large angles (30° – 45°), the SOEMLA maintains uniformly low aberrations across all angles, confirming effective suppression of angular-dependent aberrations in simulation.

3rd comment from the 2nd reviewer's comment

The reconstruction results are impressive, but the “Camera calibration and wide FOV image reconstruction” section remains brief and partly overlaps with the Methods. Please expand the description of the processing steps:

Outline the algorithmic sequence (distortion correction → homography alignment → Poisson blending).

State which software tools were used (e.g., MATLAB, Python + OpenCV) and whether GPU acceleration was employed.

Provide the approximate computation time for producing a final 1 MP stitched image from raw data and specify the hardware used. A short runtime comparison with existing compound-eye or MLA reconstruction pipelines would help readers assess practical feasibility.

Authors' response:

We thank the reviewer for this helpful suggestion. Following the request, we expanded the “Camera calibration and wide FOV image reconstruction” section to include (i) a clear outline of the algorithmic sequence with the main figure (Fig. 4a) and Methods, (ii) the software environment used, and (iii) the approximate computation time for producing a 1 MP stitched image together with the corresponding hardware specifications (Table S2). These revisions improve the clarity and transparency of the reconstruction procedure.

Page	Page 11
Original	Seamless wide FOV imaging with the SOEMLA camera was achieved through a combination of MLAs calibration and multi-view stitching (Fig. 4a). The calibration process involves a multistep framework of lens-level correction and spatial registration. First, lens shading correction was applied to each partial image using pixel-wise gain factors derived from flat-field white images, compensating for the radial intensity fall-off specific to each optical unit. Radial and tangential distortions were then corrected by estimating distortion coefficients from checkerboard calibration images. ⁴⁵ Partial images were spatially aligned using an ArUco marker array designed for the SOEMLA's layout. Corner points of detected markers were extracted using OpenCV's built-in functions, and the resulting homography matrix provides accurate registration across all views. Calibration was performed once prior to image acquisition, and the resulting parameters were reused for subsequent reconstructions without additional recalibration. Following calibration, multi-view stitching was performed using homography-based alignment and weighted blending. Each partial image was geometrically aligned using the precomputed warping matrix, and Poisson blending was applied to suppress gradient discontinuities along boundaries, yielding seamless integration. ⁴⁶ See Methods for further details.
Change	Seamless wide FOV imaging with the SOEMLA camera is achieved through a combination of an MLAs calibration stage (step I-IV) and an image reconstruction stage (step V-VI) (Fig. 4a). The calibration stage proceeds in the order of microlens region-of-interest (ROI) detection, lens shading correction, distortion correction, and homography calculation. The reconstruction stage then uses the

resulting calibration parameters to perform homography-based stitching and weighted blending. In more detail, the calibration process begins with the automatic detection of each microlens's effective ROI from a white reference frame. Next, lens shading correction is applied to each partial image using pixel-wise gain factors derived from flat-field white images, compensating for the radial intensity fall-off specific to each optical unit. Radial and tangential distortions are then corrected by estimating distortion coefficients from checkerboard calibration images.⁴⁵ Finally, partial images are spatially aligned using an ArUco marker array designed for the SOEMLA layout. Corner points of detected markers are extracted using OpenCV's built-in functions, and the resulting homography matrix provides accurate registration across all views. Calibration is performed once prior to imaging, and the resulting parameters are reused for subsequent reconstructions without additional recalibration. Following calibration, multi-view reconstruction is performed using homography-based stitching and Poisson blending. Each partial image is geometrically aligned using the precomputed warping matrix, and Poisson blending is applied to suppress gradient discontinuities along boundaries, yielding seamless integration.⁴⁶ The entire pipeline is implemented in Python using standard image-processing libraries of OpenCV, NumPy, and SciPy. The computational performance of the pipeline is benchmarked on a workstation equipped with an AMD Ryzen 9 7950X CPU (16 Cores) and 64 GB RAM, running Windows 11. No GPU acceleration is employed; instead, the processing is optimized using CPU-based parallel processing via Python's 'multiprocessing' module. The processing time for each stage is detailed in Table S2. Further details on each algorithmic step are provided in the Methods section.

Page	Original	Change
Page 13. Fig. 4a	 The original diagram shows a sequence of three images under a 'camera calibration' header. The first image shows two white spheres. The second image shows a checkerboard pattern. The third image shows a grid of markers with red and blue circles highlighting specific points.	 The revised diagram, labeled 'a', shows a sequence of four steps under a 'camera calibration' header. Step I: ROI detection (two white spheres). Step II: lens-shading correction (two white spheres with shading). Step III: distortion correction (checkerboard). Step IV: ArUco marker matching (grid of markers with red and blue circles). Arrows indicate the flow from I to II, II to III, and III to IV.

Methods

MicroLens ROI detection and elliptical masking

An automated masking procedure isolates individual microlens regions to extract reliable partial images from the raw SOEMLA camera data. A white reference image is first captured to enhance the visibility of partial image boundaries formed by each microlens. Pixels with intensity values above a global threshold T are selected. The selected bright pixel clusters are then grouped using morphological operations, and an ellipse (ε_i) is fitted using least-squares optimization for each group:

$$\varepsilon_i: Ax^2 + Bxy + Cy^2 + Dx + Ey + F = 0.$$

Lens shading correction

Flat-field white images are acquired under uniform illumination to compensate for radial brightness fall-off due to oblique incidence and vignetting. A gain map $G(x, y)$ is computed by normalizing the flat-field response:

$$G(x, y) = \frac{I_{\text{ref}}}{I_{\text{flat}}(x, y)}.$$

Distortion correction

Geometric distortions in each partial image are corrected based on the pinhole camera model described by Zhang [45]. Let (x, y) be the distorted pixel coordinates and (x_u, y_u) be the undistorted coordinates relative to the image center. The correction model simultaneously accounts for both radial and tangential distortions:

$$\begin{aligned} x_u &= x(1 + k_1r^2 + k_2r^4 + k_3r^6) + [2p_1xy + p_2(r^2 + 2x^2)] \\ y_u &= y(1 + k_1r^2 + k_2r^4 + k_3r^6) + [p_1(r^2 + 2y^2) + 2p_2xy] \end{aligned}$$

where $r^2 = x^2 + y^2$. The five distortion parameters $(k_1, k_2, k_3, p_1, p_2)$ are estimated simultaneously with the camera's intrinsic matrix by optimizing the reprojection error from multiple views of the checkerboard calibration target using OpenCV's built-in functions.

Homography estimation and registration

An ArUco marker array is used as the reference calibration pattern. Each microlens captures a distinct perspective of the planar target. For each microlens view, visible marker corners $\{\mathbf{x}_j\}$ are detected and matched with known marker positions $\{\mathbf{x}'_j\}$ in the global coordinate system. A homography matrix $\mathbf{H}_i \in \mathbb{R}^{3 \times 3}$ is computed by solving:

$$\lambda \begin{bmatrix} x'_i \\ y'_i \\ 1 \end{bmatrix} = \mathbf{H}_i \begin{bmatrix} x_i \\ y_i \\ 1 \end{bmatrix}.$$

This transformation aligned each partial image to the global stitched frame. The reprojection error for each matched corner was calculated as: $e_j = \|\mathbf{x}'_j - \mathbf{H}_i \mathbf{x}_j\|_2$.

Poisson image blending

To create a seamless final image without visible edges from stitching, we employ Poisson image blending [46]. This technique edits an image f (the final stitched canvas) by importing a source region Ω from another image g (a warped partial image) while matching the gradients. The goal is to find an unknown function f over Ω that minimizes the difference between its gradient field ∇f and the source gradient field $v = \nabla g$. This is formulated as solving a Poisson equation with Dirichlet boundary conditions:

$$\min_f \iint_{\Omega} |\nabla f - v|^2, \text{ with } f|_{\partial\Omega} = g|_{\partial\Omega}.$$

This minimization is equivalent to solving the Poisson equation $\Delta f = \text{div}(v)$ over the domain Ω , where Δ is the Laplacian operator. For the SOEMLA camera, the guidance field v is a composite of gradients from all overlapping warped images. The Dirichlet boundary conditions ($f|_{\partial\Omega}$) are defined by the pixel values at the outermost perimeter of the entire stitched canvas (i.e., regions covered by only one partial image). The resulting large, sparse linear system is formulated as a finite-difference problem and solved numerically using ‘`scipy.sparse.linalg.spsolve`’.

Table S2 Processing time for wide FOV stitching.

Stage	Process	Avg. time per unit (sec)	Avg. time per image (sec)
camera calibration	ROI detection	0.03435	0.8938
	lens shading correction	1.494	3.913
	distortion correction	25.15	40.85
	homography matching	5.603	12.43
image reconstruction	warping (homography)	0.3759	1.585
	simple blending	-	8.568
	Poisson blending	-	46.48

4th comment from the 2nd reviewer's comment

The section “SOEMLA-based high-resolution wide FOV imaging” contains valuable data but could flow more smoothly. The first paragraph discussing Fig. 3a–b starts directly with details before presenting the key idea. Consider opening with a clear statement such as:

“By tuning the aspect ratio (AR) of the ellipsoidal microlenses, the sagittal and tangential focal planes can be balanced, effectively removing astigmatic blur at oblique incidence.”

This will help readers grasp the main finding before entering the technical discussion.

Likewise, add a short transition when moving from single-lens optimization to full-array comparison (SOEMLA vs SOMLA vs spherical MLA).

Authors' response:

We fully agree with the reviewer for the constructive suggestion. To improve the flow of the section, we revised the paragraph discussing Figure 3a–b to begin with a concise statement summarizing the key concept of astigmatism correction through the aspect ratio tuning of the ellipsoidal microlenses. This change helps readers immediately grasp the main finding before entering the technical discussion. In addition, we clarified the transition between single-unit optimization and full-array comparison by adding a connecting phrase in the following paragraph. These changes enhance the readability and logical continuity of the section.

Page	Original	Change
Page 7. Line 9	The SOEMLA camera provides high-resolution wide FOV imaging by suppressing optical aberrations and maintaining uniform image quality over a broad angular range. A SOEMLA camera with aspect ratios (ARs; minor-to-major axis length) ranging from 0.80 to 1.00 was fabricated to evaluate astigmatism correction at oblique incidence (Fig. 3a).	The SOEMLA camera provides high-resolution wide FOV imaging by suppressing optical aberrations and maintaining uniform image quality over a broad angular range. In particular, tuning the aspect ratio (AR; minor-to-major axis length) of the ellipsoidal microlenses balances the sagittal and tangential focal planes and minimizes astigmatic blur at oblique incidence. A SOEMLA camera with ARs ranging from 0.80 to 1.00 was fabricated to evaluate astigmatism correction at oblique incidence (Fig. 3a).
Page 8. Line 2	The SOEMLA camera corrects astigmatism effectively at various viewing angles by tailoring the curvature of ellipsoidal microlenses (Fig. S9 and Fig. S10).	The SOEMLA camera corrects astigmatism effectively at various viewing angles by extending the single-unit optimization through tailored curvature of ellipsoidal microlenses (Fig. S9).

5th comment from the 2nd reviewer's comment

Typos:

Introduction, paragraph 3: “featuers” → “features”.

Paragraph 2: remove the comma in “...extended depth-of-focus microlenses, 28 or additional lens sets 29, 30” or rephrase as “...extended-depth-of-focus microlenses 28 or additional lens sets 29, 30.”

Keep consistent use of “wide field-of-view (FOV)” or “wide-FOV.”

Standardize units (μm^2 , $^\circ$) and ensure that all figure scale bars include units.

Authors' response:

We thank the reviewer for these helpful editorial suggestions. The typographical errors have been corrected through the main text and supplementary materials. We have also standardized terminology to ensure consistent use of “wide field-of-view (FOV)” throughout the manuscript and unified all unit formats (μm^2 , $^\circ$) while verifying that every figure scale bar includes proper units. All points have been carefully revised as suggested, significantly improving the readability and professional presentation of the manuscript.

Page	Original	Change
Page 3. Line 11	The parasitic insect Xenos peckii featuers a unique visual system of dozens of eyelets blending compound eyes and camera-type eye traits (Fig. 1a).	The parasitic insect Xenos peckii features a unique visual system of dozens of eyelets blending compound eyes and camera-type eye traits (Fig. 1a).
Page 14. Line 22	Dental phantom was also captured to demonstrate the high-resolution wide FOV imaging within a confined space under realistic conditions.	A dental phantom was also captured to demonstrate the high-resolution wide FOV imaging within a confined space under realistic conditions.
Page 17. Line 22	Direction-specific spatial offsets and asymmetric microlens curvatures allows clear angular sampling with sharp PSFs at oblique angles by effectively reducing optical aberrations.	Direction-specific spatial offsets and asymmetric microlens curvatures allow clear angular sampling with sharp PSFs at oblique angles by effectively reducing optical aberrations.
Page 17. Line 30	Fully stitched images demonstrate high-resolution wide FOV imaging at shot distances for real-world targets such as microfluidic chip, dental phantom, and human face.	Fully stitched images demonstrate high-resolution wide FOV imaging at shot distances for real-world targets such as a microfluidic chip, a dental phantom, and a human face.
Figure S4	Elliptical microcylinders with axis dimensions (a_1 , a_2) and thickness t reflow into ellipsoidal microlenses at 150oC. The sag height remains	Elliptical microcylinders with axis dimensions (a_1, a_2) and thickness t reflow into ellipsoidal microlenses at 150°C. The sag height remains

	constant ($h \approx 2t$) while ROC along each axis is determined by the corresponding axis length by a_i .	constant ($h \approx 2t$) while ROC along each axis is determined by the corresponding axis length by a_i .
Fig. S1	(a) Schematic illustration of a single SOA unit. The upper aperture functions as the aperture stop (AS) that controls the light intensity while the lower aperture acts as the field stop (FS) that limits the acceptance angle.	(a) Schematic illustration of a single SOA unit. The upper aperture functions as the aperture stop (AS) that controls the light intensity while the lower aperture acts as the field stop (FS) that limits the acceptance angle.
Fig. S5	(c) Design parameters of the ellipsoidal microlens curvature.	(c) Design parameters of the ellipsoidal microlens curvature.
Fig. S6	Other aberrations such as coma and spherical aberration are excluded from this analysis, as their contributions are negligible compared to dominant FC and ACT at large incidence angles with sufficiently small aperture sizes.	Other aberrations such as coma and spherical aberration are excluded from this analysis, as their contributions are negligible compared to dominant FC and AST at large incidence angles with sufficiently small aperture sizes.
Fig. S6	Asymmetric convergence of sagittal (red) and tangential (blue) rays creates two distinct focal planes, z_S and z_T , leading to astigmatic blur.	Asymmetric convergence of sagittal (red) and tangential (blue) rays creates two distinct focal planes, z_S and z_T , leading to astigmatic blur.
Fig. S7	Top view of the fully assembled SOEMLA camera showing the upper apertures, gap spacers, and CMOS ISA.	Top view of the fully assembled SOEMLA camera showing the upper apertures, gap spacers, and CMOS ISA. Scale bar: 2 mm.
Fig. S8	Ellipsoidal microlens with independently optimized sagittal ($R_x = 472 \mu\text{m}$) and tangential ($R_y = 1054 \mu\text{m}$) curvatures.	Ellipsoidal microlens with independently optimized sagittal ($R_x = 472 \mu\text{m}$) and tangential ($R_y = 1054 \mu\text{m}$) curvatures.
Fig. S10	Simulated PSFs and images of (a) a spherical microlens and (b) an ellipsoidal microlens under 45° incidence. The spherical microlens shows sagittal elongation of the PSF (RMS radius: $15.5 \mu\text{m}$) due to astigmatism, resulting in blurring and low image contrast (0.2). In contrast, the ellipsoidal microlens tailored for 45° incidence produces a symmetric PSF (RMS radius: $2.61 \mu\text{m}$) and restores image contrast to 1.0.	Simulated PSFs and images of (a) a spherical microlens and (b) an ellipsoidal microlens under 45° incidence. The spherical microlens shows sagittal elongation of the PSF (RMS radius: $15.5 \mu\text{m}$) due to astigmatism, resulting in blurring and low image contrast (0.2). In contrast, the ellipsoidal microlens tailored for 45° incidence produces a symmetric PSF (RMS radius: $2.61 \mu\text{m}$) and restores image contrast to 1.0. Scale bar: $10 \mu\text{m}$.

6th comment from the 2nd reviewer’s comment

The supplementary figures are well prepared. Adding a simple schematic summarizing the complete optical stack (microlens + SOA + sensor) would help visualize layer functions and offsets. In Fig. S10, consider showing the RMS spot-radius versus incidence-angle comparison directly in the main text or as an inset, to emphasize the optical benefit of the SOEMLA design.

Authors’ response:

We appreciate the reviewer’s constructive suggestion. In response to the request for a clearer visualization of the complete optical stack, we have added a high-resolution optical cross-section image (Figure S7b) with color overlays indicating the precise positions of the microlens, spatially offset apertures (SOAs), and the sensor. This figure and its detailed caption provide a clear visual summary of the fabricated SOEMLA structure and the function of the different layers.

Page	Original	Change
Fig. S7	 (a) glass substrate ellipsoidal microlenses gap spacers CMOS ISA	 (b) chief rays apertures glass substrate microlenses spacer active pixel area PCB
Fig. S7	(b) Side view illustrating the vertically integrated optical stack, which includes ellipsoidal microlenses, gap spacers, and the CMOS ISA.	(b) Cross-sectional optical image of the microfabricated SOEMLAs integrated with the CMOS ISA. The upper and lower apertures (120 nm thick) are not visible in the optical image and are indicated by color overlays to denote the positions. As the incidence angle increases, the spatial offset and microlens size increase correspondingly to maintain directional imaging. Scale bar: 500 μm.

Furthermore, the RMS spot-radius analysis derived from Zemax optical simulations is presented in Fig. S10 to quantify the theoretical imaging performance of the SOEMLA camera design. In the main text, we focused on the experimentally measured results to represent the realized optical performance. To avoid confusion between simulation-based and experimental results, the RMS spot-radius comparison remains in the Supplementary Information. However, a concise summary of the Fig. S10 findings has been incorporated into the revised manuscript to emphasize the SOEMLA’s aberration-correction capability.

Page	Original	Change
Page 8. Line 3	The SOEMLA camera corrects astigmatism effectively at various viewing angles by tailoring the curvature of ellipsoidal microlenses (Fig. S9 and Fig. S10).	The SOEMLA camera corrects astigmatism effectively at various viewing angles by extending the single-unit optimization through tailored curvature of ellipsoidal microlenses (Fig. S9). Ray-tracing results further show that the SOEMLA configuration maintains a nearly constant RMS spot radius across wide incidence angles, confirming superior aberration correction capability (Fig. S10-11).

For the 3rd Reviewer's comments
General comments and recommendation

*The submitted manuscript describes an interesting further development of a microlens array camera using spatially-displaced elliptical microlenses to strongly reduce the aberrations of off-axis components of the image and thereby significantly increase the usable field-of-view for a planar imager. The concept is inspired by the ocular system of *xenos peckii*, on which numerous imaging systems developed by this research group have been based, and is part of interesting ongoing research into artificial compound eye cameras.*

Whereas microlens arrays coupled with aperture arrays have been widely demonstrated for segmented imaging applications, to my knowledge the use of elliptical lenses to reduce aberrations and thus increase resolution has not been shown before. As such, the idea presented here is novel and represents a step forward in the improvement of these "nature inspired" compound eye cameras. It should be noted that the authors previously published this concept as conference proceedings (DOE 10.1109/OMN61224.2024.10685283 and 10.1109/OMN65869.2025.11125932), but less comprehensively than in the present submission; these publications are not cited in the paper.

The paper is generally well written and clearly structured, with good illustrations and adequate presentation of the data. The conclusions are well supported and the improvement in imaging performance is such that the new approach represents a technical advance over previous competing systems. In addition, a few comments concerning some details in the text:

(NB: page and/or line numbers would have been really helpful 😊)

Authors' response:

We sincerely thank the reviewer for the positive and constructive evaluation of our manuscript. We appreciate the reviewer's recognition of the SOEMLA camera's novelty and technical merit, as well as the thoughtful suggestions to further clarify the optical configuration and design rationale. In response, we have refined the main text and expanded the Supplementary Information (Fig. S11, Fig. S16, Supplementary Note 1) with additional quantitative and theoretical analyses to better support the key claims of the study. We believe that these revisions directly address the reviewer's concerns and have substantially improved the overall clarity, completeness, and presentation quality of the manuscript.

The two mentioned conference papers (DOE 10.1109/OMN61224.2024.10685283 and 10.1109/OMN65869.2025.11125932) presented preliminary and highly abbreviated summaries of early-stage prototypes without detailed theory, fabrication, or experimental validation. According to common publication practices, such preliminary conference extended abstracts are generally not cited in the subsequent full journal article when the journal version entirely supersedes the earlier work. We confirm that the current manuscript is the first complete and peer-reviewed presentation of the full optical design rationale, wafer-scale fabrication, comprehensive characterization, and wide FOV imaging results. The conference summaries do not contain any unique methods, results, or

interpretations that require citation for context or attribution. Nevertheless, we appreciate the reviewer’s attention to this point and assure that there is no overlap beyond the allowable preliminary disclosure typically permitted by both conferences and journals.

Lastly, we also apologize for the absence of page and line numbers in the original submission, which have been added in the revised version for the reviewer’s convenience.

Specific comments

1st comment from the 3rd reviewer’s comment

3rd paragraph – “Like Xenos peckii’s eyelet, each optical unit contains SOAs, an ellipsoidal microlens, and sub-active pixels of CMOS”: I do not believe that xenos peckii has elliptical eyelets. Also, what are “sub-active pixels”? Do you mean “subapertures”? Or are you referring to the active pixel sensors on the CMOS chip?

Authors’ response:

We agree that *Xenos peckii* does not possess elliptical eyelets. The biological analogy refers only to its directionally separated visual sampling behavior (chunk sampling – unlike point sampling of general compound eyes), which inspired the spatially offset architecture of our optical units. The ellipsoidal microlenses are an engineering design introduced independently to compensate for aberrations at oblique incidence when using planar CMOS ISA. In addition, the term “sub-active pixels” has been revised to “a designated pixel region on the CMOS ISA” to more clearly indicate the portion of sensor pixels assigned to each optical unit.

Page	Original	Change
Page 3. Line 16	Like Xenos peckii ’s eyelet, each optical unit contains SOAs, an ellipsoidal microlens, and sub-active pixels of CMOS ISA, which captures a separate view from a particular direction.	Like Xenos peckii ’s visual sampling , each optical unit contains SOAs, an ellipsoidal microlens, and a designated pixel region on the CMOS ISA , which captures a separate view from a particular direction.

2nd comment from the 3rd reviewer's comment

A few sentences further – “angular sensitivity”: I think this is misleading, I imagine you mean something like “the angle range which is imaged” or “visual axis angle”.

Authors' response:

To clarify, the sentence has been revised to use “angular range”, which more accurately describes the geometric field angle defined by the spatial offset between the upper and lower apertures.

Page	Original	Change
Page 3. Line 20	In particular, the SOAs consist of upper and lower apertures that are laterally displaced by a spatial offset to define the angular sensitivity of each optical unit.	In particular, the SOAs consist of upper and lower apertures that are laterally displaced by a spatial offset to define the angular range of each optical unit.

3rd comment from the 3rd reviewer's comment

Section Results, 1st paragraph – I would prefer to see something like figure S11 in the main paper, perhaps combined with a larger version of Fig. 2b; otherwise the overall structure, and the novelty of the displaced and rotated elliptical microlenses, remains rather unclear for the reader who might not look at the supplementary material.

Authors' response:

We appreciate the reviewer for this thoughtful suggestion. We fully agree that the structural configuration and novelty of the displaced and rotated ellipsoidal microlenses are central to this work and must be clearly conveyed to all readers. While we recognize the value of moving Fig. S11 to the main text, its content would largely overlap with the existing optical image shown in Fig. 2b and the detailed configuration already presented in Fig. S5. Instead, we have revised the corresponding image and paragraph in the Results section to explicitly describe the radial arrangement, spatial offset, and rotation of the 5×7 ellipsoidal microlenses. This ensures that the key structural concept of the SOEMLA camera can be fully understood without referring to the Supplementary Information.

Page	Original	Change
Page 5. Line 20	The optical image clearly demonstrates that the ellipsoidal microlenses are radially arranged with spatially varying dimensions (Fig. 2b). In particular, the peripheral microlenses exhibit large aspect ratios and lens sizes to capture wide viewing angles (Fig. S5).	The optical image clearly demonstrates that the ellipsoidal microlenses are radially arranged with spatially varying dimensions and orientations, corresponding to the displaced and rotated 5×7 configurations designed to extend the overall FOV (Fig. 2b). In particular, the peripheral microlenses exhibit large aspect ratios and lens sizes to capture wide viewing angles (detailed parameters of each unit are provided in Fig. S5).
Page 6. Fig. 2	(b) Optical image of the microfabricated SOEMLAs. The ellipsoidal microlenses feature radially arranged curvatures with variable spatial offsets to reduce astigmatism and field curvature in wide FOV imaging. Scale bar: 1 mm.	(b) Optical image of the microfabricated SOEMLAs with variable sizes and asymmetry . The ellipsoidal microlenses feature radially arranged curvatures with variable spatial offsets to reduce astigmatism and field curvature in wide FOV imaging. Scale bar: 1 mm.
Page 6. Fig. 2		
4th comment from the 3rd reviewer's comment

A few sentences further – “The measured profile closely matched the target design ... showing minimal astigmatism and field curvature along both the X–X and Y–Y cross-sections.”: the lens profile cannot show minimal astigmatism or field curvature, only the optical field passing through that lens profile can.

Also, whereas astigmatism is clear, how is field curvature characterized in your simulations and in the measurements? Distance of focus from the focal plane of the imager? How do you substantiate the assertion that field curvature has been minimized?

Authors' response:

We appreciate the reviewer's clarification. As noted, the lens curvature itself does not directly demonstrate aberration reduction. Our original intention was to convey that the designed curvature for minimizing aberrations was accurately fabricated, and we have revised the sentence accordingly to

reflect this intent. The quantitative analysis of aberration reduction and the resulting resolution improvement is presented in Fig. 3.

Furthermore, to clarify the quantitative characterization of field curvature and its distinction from astigmatism, we have added Fig. S11, which presents the calculated Seidel aberration coefficients (S_1 – S_4) for both the spherical MLA and SOEMLA cameras as a function of incidence angle. This analysis allows readers to quantitatively compare how major aberrations vary with incidence angle and how they are mitigated through the SOEMLA's design. In the SOEMLA camera, the gradual increase in curvature from the center to the periphery compensates for field curvature, while the introduced surface asymmetry minimizes astigmatism as described in the main text (Page 5 Line 30 – Page 6 Line 2).

Page	Original	Change
Page 5. Line 27	The measured profile closely matched the target design from Zemax OpticStudio (Ansys Inc.), showing minimal astigmatism and field curvature along both the X–X' and Y–Y' cross-sections.	The measured profile closely matches the Zemax OpticStudio (Ansys Inc.) target design, minimizing astigmatism and field curvature along both the X–X' and Y–Y' cross-sections.
Page 8. Line 3	The SOEMLA camera corrects astigmatism effectively at various viewing angles by tailoring the curvature of ellipsoidal microlenses (Fig. S9 and Fig. S10).	The SOEMLA camera corrects astigmatism effectively at various viewing angles by extending the single-unit optimization through tailored curvature of ellipsoidal microlenses (Fig. S9). Ray-tracing results further show that the SOEMLA configuration maintains a nearly constant RMS spot radius across wide incidence angles, confirming superior aberration correction capability (Fig. S10-11).

Fig. S11 Calculated Seidel aberration coefficients (S_1 – S_4) as a function of incidence angle for the spherical MLA and SOEMLA cameras. The coefficients correspond to S_1 spherical aberration, S_2 coma, S_3 astigmatism, and S_4 field curvature. While the spherical MLA exhibits rapidly increasing astigmatism and field curvature at large angles (30° – 45°), the SOEMLA maintains uniformly low aberrations across all angles, confirming effective suppression of angular-dependent aberrations in simulation.

5th comment from the 3rd reviewer's comment

Section SOEMLA-based high-resolution wide FOV imaging, 1st paragraph, discussion of the 90AR microlens: you should emphasize that this is one example microlens, since the other profiles have a different visual axis and different characteristics.

Authors' response:

We appreciate the reviewer's helpful suggestion. We agree that the 0.90-AR microlens should be described as a representative example, as other aspect ratios exhibit different visual axes and optical characteristics. To clarify this point, we have added a note indicating that the 0.90-AR microlens corresponds to a 20° visual axis and that both curvature and asymmetry can be tuned across the wide FOV depending on the incidence angle.

Page	Original	Change
Page 7. Line 30	The experimental results demonstrate that ellipsoidal microlenses effectively reduce angular-dependent astigmatism without relying on complex optical assemblies or active control.	The experimental results demonstrate that ellipsoidal microlenses effectively reduce angular-dependent astigmatism without relying on complex optical assemblies or active control. Note that the 0.90-AR microlens corresponds to a 20° visual axis, while each optical channel can be tuned in curvature and asymmetry across the full FOV to match local incidence angles.

6th & 7th comments from the 3rd reviewer's comment

Same section, 2nd paragraph, 5th sentence: I think you mean SOMLA, not SOEMLA.

Same section, last paragraph: please define TTL.

Authors' response:

We sincerely thank the reviewer for catching these important typographical and clarity issues. The term has been corrected from SOEMLA to SOMLA, and the abbreviation TTL has been defined as total track length in the revised manuscript for clarity.

Page	Original	Change
Page 8. Line 12	The SOEMLA camera eliminates field curvature by adjusting the focal length of each microlens to the designated viewing angle.	The SOMLA camera eliminates field curvature by adjusting the focal length of each microlens to the designated viewing angle.
Page 8. Line 23	The SOEMLA camera achieves 0.94 mm TTL and 140° FOV with a 7.9 mm sensor, whereas the commercial camera (Camera Module 3 Wide, Raspberry Pi Ltd.) has 8.3 mm TTL and 120° FOV with a 7.4 mm sensor.	The SOEMLA camera achieves 0.94 mm total track length (TTL) and 140° FOV with a 7.9 mm sensor, whereas the commercial camera (Camera Module 3 Wide, Raspberry Pi Ltd.) has 8.3 mm TTL and 120° FOV with a 7.4 mm sensor.

8th comment from the 3rd reviewer's comment

Same paragraph and Fig. 3f: it is not clear why your camera should have infinite DoF. Please elaborate on this.

Authors' response:

We appreciate the reviewer for this insightful comment regarding the origin of the effectively infinite depth of field (DOF) in the SOEMLA camera. To clarify this point, we have elaborated on the underlying optical mechanism and added Supplementary Note 1, which provides a quantitative calculation of the hyperfocal distance.

As detailed in the new note, the SOEMLA camera exhibits a hyperfocal distance shorter than 10 mm ($H \leq 9.93$ mm), meaning that all objects located beyond approximately 5 mm remain in focus, thereby achieving an effectively infinite DOF. This is due to the extremely short focal length of the individual microlenses

This quantitative analysis clarifies the optical basis of the effectively infinite DOF and has been incorporated into the revised manuscript.

Page	Original	Change
Page 9. Line 2	In contrast, the SOEMLA camera decouples depth-dependent focal variation from angular aberrations, ensuring consistent MTF50 values across both depth and angle. Along with high resolution and infinite DOF, the SOEMLA camera clearly demonstrates proximal imaging (< 50 mm), thereby reducing the overall optical path (WD + TTL) and facilitating further miniaturization. T	In contrast, the SOEMLA camera decouples depth-dependent focal variation from angular aberrations, ensuring consistent MTF50 values across both depth and angle. The SOEMLA camera exhibits a hyperfocal distance shorter than 10 mm, resulting in effectively infinite DOF (Supplementary Note 1). Unlike conventional lens systems that extend DOF at the cost of brightness, the SOEMLA camera maintains overall light collection efficiency through its inherently short focal length. Measured signal-to-noise ratio (SNR) demonstrates practical photon throughput across visual-axis angles (Fig. S12–S13). Along with high resolution and infinite DOF, the SOEMLA camera clearly demonstrates proximal imaging (< 50 mm), thereby reducing the overall optical path (WD + TTL) and facilitating further miniaturization.
Note 1	Supplementary Note 1. Hyperfocal distances of the SOEMLA camera The hyperfocal distance (H) defines the closest distance at which a lens can be focused while keeping objects at infinity acceptably sharp. When the image sensor is focused at H, the depth-of-field (DOF) extends from $H/2$ to infinity, and when H becomes shorter than the typical object distance, the system effectively exhibits an infinite DOF. The hyperfocal distance is given by $H = \frac{f^2}{Nc} + f,$ where f is the focal length of the microlens, N is the f-number, and c is the permissible circle of confusion (CoC). Adopting a diffraction-limited criterion for visible light, we set the CoC to the Airy-disk diameter $c = 2.44 \lambda N$ (with $\lambda = 550$ nm), which yields $H \leq 9.93$ mm based on the SOEMLAs' focal length and f-numbers. This result indicates that the SOEMLA camera maintains focus from approximately 5 mm to infinity, thereby achieving an effectively infinite DOF across the entire FOV.	

9th comment from the 3rd reviewer's comment

Section Camera calibration and wide FOV image reconstruction, last paragraph: how was SSIM calculated?

Authors' response:

We thank the reviewer for pointing out the need for clarification regarding the PSNR/SSIM calculation procedure. As described in the revised manuscript, the ground-truth images used for quantitative comparison were the original photographs displayed on the LCD screen. These reference images were geometrically aligned with the reconstructed results through rotation and resizing to compensate for positional misalignment and resolution differences. The structural similarity index (SSIM) and peak signal-to-noise ratio (PSNR) were then computed between the aligned image pairs, as now described in the revised text. This clarification ensures reproducibility of the quantitative image quality analysis.

Page	Original	Change
Page 12. Line 21	Quantitative image quality was evaluated by comparing the reconstructed images with the ground truth using peak signal-to-noise ratio (PSNR) and structural similarity index (SSIM). The PSNR values are 21.46 dB for the wasp target and 18.72 dB for the butterfly target with corresponding SSIM values of 0.799 and 0.786.	Quantitative image quality was evaluated by comparing the reconstructed images with the ground truth using peak signal-to-noise ratio (PSNR) and structural similarity index (SSIM). In the experiment, the ground truth images were the original photographs displayed on the LCD. They were aligned to the reconstructed images via rotation and resizing to correct positional and resolution mismatches. The PSNR values are 21.46 dB for the wasp target and 18.72 dB for the butterfly target with corresponding SSIM values of 0.799 and 0.786.

10th comment from the 3rd reviewer's comment

Section Real-world demonstration of high-resolution wide FOV imaging: you compare the performance of your SOEMLA with a spherical MLA camera. Please describe the characteristics of this camera, to allow a fair comparison. Also, what computational steps did you take to generate the images with the spherical MLA camera? For example, you show significant vignetting for the spherical MLA: why not do “lens shading correction”, as you do for the SOEMLA, for this camera as well?

Authors' response:

We appreciate the reviewer's insightful comment regarding the characteristics and image-processing conditions of the spherical MLA camera used for comparison. As described in the revised manuscript, the spherical MLA camera was fabricated with the same optical parameters as the central optical unit of the SOEMLA but employed single-layer apertures. This structural difference isolates the effect of the offset and ellipsoidal design elements for fair comparison. For the analysis in Fig. 3, both systems were evaluated under identical optical and computational conditions without applying any additional correction such as lens shading, ensuring a direct comparison of their intrinsic optical performance. This approach allows a fair comparison without introducing computational bias. For the large-area microfluidic imaging, identical image-processing steps were applied to the spherical MLA cameras to allow fair comparison under equivalent post-processing conditions in Fig. S16.

Page	Original	Change
Page 8. Line 9	The spatial resolution of the SOEMLA camera was evaluated with respect to the incidence angle and compared with other MLA cameras (Fig. 3c). Conventional MLA cameras using spherical microlenses with single-layer apertures show significant resolution degradation at large incidence angles due to both optical field curvature and astigmatism (blue).	The spatial resolution of the SOEMLA camera was evaluated with respect to the incidence angle and compared with other MLA cameras (Fig. 3c). Conventional MLA cameras using spherical microlenses with single-layer apertures show significant resolution degradation at large incidence angles due to both optical field curvature and astigmatism (blue). For a fair comparison, the spherical MLA camera was fabricated with the same optical specifications as the central optical unit of the SOEMLA but employed single-layer apertures ($f = 440 \mu\text{m}$, $F/3$).
Page 14. Line 11	In contrast, the spherical MLA camera showed a 40 mm FOV with significant vignetting and peripheral contrast loss from optical aberrations (Fig. 5c).	In contrast, the spherical MLA camera showed a 40 mm FOV with significant vignetting and peripheral contrast loss from optical aberrations using the same spherical MLA camera characterized in Fig. 3c. (Fig. 5c).
Page 14.	In comparison, the spherical MLA camera (line e–f in Fig. 5c) produced low-modulation profiles	In comparison, the spherical MLA camera (line e–f in Fig. 5c) produced low-modulation profiles

Line 19	and failed to resolve fine structures due to reduced resolution.	and failed to resolve fine structures due to reduced resolution. The lens shading- and distortion-corrected result for the spherical MLA camera is shown in Fig. S16.
---------	--	--